# On Embeddings for Numerical Features
# in Tabular Deep Learning

**Yury Gorishniy**[*]
Yandex

**Ivan Rubachev**
HSE, Yandex

**Artem Babenko**
Yandex

## Abstract

Recently, Transformer-like deep architectures have shown strong performance on tabular data problems. Unlike traditional models, e.g., MLP, these architectures map scalar values of numerical features to high-dimensional embeddings before mixing them in the main backbone. In this work, we argue that embeddings for numerical features are an underexplored degree of freedom in tabular DL, which allows constructing more powerful DL models and competing with gradient boosted decision trees (GBDT) on some GBDT-friendly benchmarks (that is, where GBDT outperforms conventional DL models). We start by describing two conceptually different approaches to building embedding modules: the first one is based on a piecewise linear encoding of scalar values, and the second one utilizes periodic activations. Then, we empirically demonstrate that these two approaches can lead to significant performance boosts compared to the embeddings based on conventional blocks such as linear layers and ReLU activations. Importantly, we also show that embedding numerical features is beneficial for many backbones, not only for Transformers. Specifically, after proper embeddings, simple MLP-like models can perform on par with the attention-based architectures. Overall, we highlight embeddings for numerical features as an important design aspect with good potential for further improvements in tabular DL. The source code is available at `https://github.com/Yura52/tabular-dl-num-embeddings`.

## 1 Introduction

Tabular data problems are currently a final frontier for deep learning (DL) research. While the most recent breakthroughs in NLP, vision, and speech are achieved by deep models [12], their success in the tabular domain is not convincing yet. Despite a large number of proposed architectures for tabular DL [2, 3, 13, 17, 21, 24, 31, 39, 40], the performance gap between them and the "shallow" ensembles of decision trees, like GBDT, often remains significant [13, 36].

The recent line of works [13, 24, 39] reduce this performance gap by successfully adapting the Transformer architecture [45] for the tabular domain. Compared to traditional models, like MLP or ResNet, the proposed Transformer-like architectures have a specific way to handle numerical features of the data. Namely, they map scalar values of numerical features to high-dimensional embedding vectors, which are then mixed by the self-attention modules. Beyond transformers, mapping numerical features to vectors was also employed in different forms in the click-through rate (CTR) prediction problems [8, 14, 40]. Nevertheless, the literature is mostly focused on developing more powerful backbones while keeping the design of embedding modules relatively simple. In particular, the existing architectures [13, 14, 24, 39, 40] construct embeddings for numerical features using quite restrictive parametric mappings, e.g., linear functions, which can lead to suboptimal performance. In this work, we demonstrate that the embedding step has a substantial impact on the model effectiveness, and its proper design can significantly improve tabular DL models.

---

[*]Correspondence to: `yuragorishniy@icloud.com`

36th Conference on Neural Information Processing Systems (NeurIPS 2022).

Specifically, we describe two different building blocks suitable for constructing embeddings for numerical features. The first one is a piecewise linear encoding that produces alternative initial representations for the original scalar values and is based on feature binning, a long-existing preprocessing technique [11]. The second one relies on periodic activation functions, which is inspired by their usage in implicit neural representations [28, 38, 42], NLP [41, 45] and CV tasks [25]. The first approach is simple, interpretable and non-differentiable, while the second demonstrates better results on average. We observe that DL models equipped with our embedding schemes successfully compete with GBDT on GBDT-friendly benchmarks and achieve the new state-of-the-art on tabular DL.

As another important finding, we demonstrate that the step of embedding the numerical features is universally beneficial for different deep architectures, not only for Transformer-like ones. In particular, we show, that after proper embeddings, simple MLP-like architectures often provide the performance comparable to the state-of-the-art attention-based models. Overall, our work demonstrates the large impact of the embeddings of numerical features on the tabular DL performance and shows the potential of investigating more advanced embedding schemes in future research.

To sum up, our contributions are as follows:

1. We demonstrate that embedding schemes for numerical features are an underexplored research question in tabular DL. Namely, we show that more expressive embedding schemes can provide substantial performance improvements over prior models.
2. We show that the profit from embedding numerical features is not specific for Transformer-like architectures, and proper embedding schemes benefit traditional models as well.
3. On a number of public benchmarks, we achieve the new state-of-the-art on tabular DL.

## 2   Related work

**Tabular deep learning.** During several recent years, the community has proposed a large number of deep models for tabular data [2, 3, 13, 15, 17, 21, 24, 31, 39, 40, 46]. However, when systematically evaluated, these models do not consistently outperform the ensembles of decision trees, such as GBDT (Gradient Boosting Decision Tree) [7, 19, 32], which are typically the top-choice in various ML competitions [13, 36]. Moreover, several recent works have shown that the proposed sophisticated architectures are not superior to properly tuned simple models, like MLP and ResNet [13, 18]. In this work, unlike the prior literature, we do not aim to propose a new backbone architecture. Instead, we focus on more accurate ways to handle numerical features, and our developments can be potentially combined with any model, including traditional MLPs and more recent Transformer-like ones.

**Transformers in tabular DL.** Due to the tremendous success of Transformers for different domains [10, 45], several recent works adapt their self-attention design for tabular DL as well [13, 17, 24, 39]. Compared to existing alternatives, applying self-attention modules to the numerical features of tabular data requires mapping the scalar values of these features to high-dimensional embedding vectors. So far, the existing architectures perform this "scalar" $\rightarrow$ "vector" mapping by relatively simple computational blocks, which, in practice, can limit the model expressiveness. For instance, the recent FT-Transformer architecture [13] employs only a single linear layer. In our experiments, we demonstrate that such embedding schemes can provide suboptimal performance, and more advanced schemes often lead to substantial profit.

**CTR Prediction.** In CTR prediction problems, objects are represented by numerical and categorical features, which makes this field highly relevant to tabular data problems. In several works, numerical features are handled in some non-trivial way while not being the central part of the research [8, 40]. Recently, however, a more advanced scheme has been proposed in Guo et al. [14]. Nevertheless, it is still based on linear layers and conventional activation functions, which we found to be suboptimal in our evaluation.

**Feature binning.** Binning is a discretization technique that converts numerical features to categorical features. Namely, for a given feature, its value range is split into bins (intervals), after which the original feature values are replaced with discrete descriptors (e.g. bin indices or one-hot vectors) of the corresponding bins. We point to the work by Dougherty et al. [11], which performs an overview of some classic approaches to binning and can serve as an entry point to the relevant literature on the topic. In our work, however, we utilize bins in a different way. Specifically, we use their edges to construct lossless piecewise linear representations of the original scalar values. It turns out that this

simple and interpretable representations can provide substantial benefit to deep models on several tabular problems.

**Periodic activations.** Recently, periodic activation functions have become a key component in processing coordinates-like inputs, which is required in many applications. Examples include NLP [45], CV [25], implicit neural representations [28, 38, 42]. In our work, we show that periodic activations can be used to construct powerful embedding modules for numerical features in tabular data problems. Contrary to some of the aforementioned papers, where components of the multidimensional coordinates are mixed (e.g. with linear layers) before passing them to periodic functions [38, 42], we find it crucial to embed each feature separately before mixing them in the main backbone.

## 3 Embeddings for numerical features

In this section, we describe the general framework for what we call "embeddings for numerical features" and the main building blocks used in the experimental comparison in section 4.

**Notation.** For a given supervised learning problem on tabular data, we denote the dataset as $\left\{\left(x^j, y^j\right)\right\}_{j=1}^n$ where $y^j \in \mathbb{Y}$ represents the object's label and $x^j = \left(x^{j(num)}, x^{j(cat)}\right) \in \mathbb{X}$ represents the object's features (numerical and categorical). $x_i^{j(num)}$, in turn, denotes the $i$-th numerical feature of the $j$-th object. Depending on the context, the $j$ index can be omitted. The dataset is split into three disjoint parts: $\overline{1, n} = J_{train} \cup J_{val} \cup J_{test}$, where the "train" part is used for training, the "validation" part is used for early stopping and hyperparameter tuning, and the "test" part is used for the final evaluation.

### 3.1 General framework

We formalize the notion of "embeddings for numerical features" as $z_i = f_i((x_i^{(num)}) \in \mathbb{R}^{d_i}$, where $f_i(x)$ is the embedding function for the $i$-th numerical feature, $z_i$ is the embedding of the $i$-th numerical feature and $d_i$ is the dimensionality of the embedding. Importantly, the proposed framework implies that embeddings for all features are computed *independently* of each other. Note that the function $f_i$ can depend on parameters that are trained as a part of the whole model or in some other fashion (e.g. before the main optimization). In this work, we consider only embedding schemes where the embedding functions for all features are of the same functional form. We never share parameters of embedding functions of different features.

The subsequent use of the embeddings depends on the model backbone. For MLP-like architectures, they are concatenated into one flat vector (see Appendix A for illustrations). For Transformer-based architectures, no extra step is performed and the embeddings are passed as is, so the usage is defined by the original architectures.

### 3.2 Piecewise linear encoding

While vanilla MLP is known to be a universal approximator [9, 16], in practice, due to optimization peculiarities, it has limitations in its learning capabilities [34]. However, the recent work by Tancik et al. [42] uncovers the case where changing the input space alleviates the above issue. This observation motivates us to check if changing the representations of the original scalar values of numerical features can improve the learning capabilities of tabular DL models.

At this point, we try to start simple and turn to "classical" machine learning techniques. Namely, we take inspiration from the one-hot encoding algorithm that is widely and successfully used for representing discrete entities such as categorical features in tabular data problems or tokens in NLP. We note that the one-hot representation can be seen as an opposite solution to the scalar representation in terms of the trade-off between parameter efficiency and expressivity. To check whether the one-hot-like approach can be beneficial for tabular DL models, we design a continuous alternative to the one-hot encoding (since the vanilla one-hot encoding is barely applicable to numerical features).

Formally, for the $i$-th numerical feature, we split its value range into the disjoint set of $T^i$ intervals $B_1^i, \ldots, B_T^i$, which we call *bins*: $B_t^i = [b_{t-1}^i, b_t^i)$. The splitting algorithm is an important implementation detail that we discuss later. From now on, we omit the feature index $i$ for simplicity. Once the bins are determined, we define the encoding scheme as in Equation 1:

$$\text{PLE}(x) = [e_1, \ldots, e_T] \in \mathbb{R}^T$$

$$e_t = \begin{cases} 0, & x < b_{t-1} \text{ AND } t > 1 \\ 1, & x \geq b_t \text{ AND } t < T \\ \frac{x - b_{t-1}}{b_t - b_{t-1}}, & \text{otherwise} \end{cases} \quad (1)$$

where PLE stands for "**p**eicewise **l**inear **e**ncoding". We provide the visualization in Figure 1.

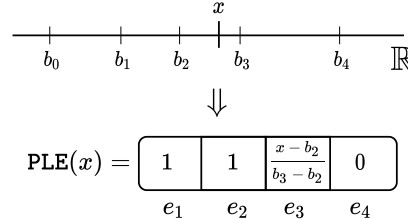

Figure 1: The piecewise linear encoding (PLE) in action for $T = 4$ (see Equation 1).

Note that:

- PLE produces alternative initial representations for the numerical features and can be viewed as a preprocessing strategy. These representations are computed once and then used instead of the original scalar values during the main optimization.
- For $T = 1$, the PLE-representation is effectively equivalent to the scalar representation.
- Contrary to categorical features, numerical features are ordered; we express that by setting to $1$ the components corresponding to bins with the right boundaries lower than the given feature value (this approach resembles how labels are encoded in ordinal regression problems).
- The cases $(x < b_0)$ and $(x \geq b_T)$ are also covered by Equation 1 (which leads to $(e_1 \leq 0)$ and $(e_T \geq 1)$ respectively).
- The choice to make the representation piecewise linear is itself a subject for discussion. We analyze some alternatives in subsection 5.2.
- PLE can be viewed as feature preprocessing, which is additionally discussed in subsection 5.3.

**A note on attention-based models.** While the described PLE-representations can be passed to MLP-like models as is, attention-based models are inherently invariant to the order of input embeddings, so one additional step is required to add the information about feature indices to the obtained encodings. Technically, we observe that it is enough to place one linear layer after PLE(without sharing weights between features). Conceptually, however, this solution has a clear semantic interpretation. Namely, it is equivalent to allocating one trainable embedding $v_t \in \mathbb{R}^d$ for each bin $B_t$ and obtaining the final feature embedding by aggregating the embeddings of its bins with $e_t$ as weights, plus bias $v_0$. Formally: $f_i(x) = v_0 + \sum_{t=1}^{T} e_t \cdot v_t = \texttt{Linear}(\text{PLE}(x))$.

In the following two sections, we describe two simple algorithms for building bins suitable for PLE. Namely, we rely on the classic binning algorithms [11] and one of the two algorithms is unsupervised, while another one utilizes labels for constructing bins.

### 3.2.1 Obtaining bins from quantiles

A natural baseline way to construct the bins for PLE is by splitting value ranges according to the uniformly chosen empirical quantiles of the corresponding individual feature distributions. Formally, for the $i$-th feature: $b_t = \mathbb{Q}_{\frac{t}{T}}\left(\{x_i^{j(num)}\}_{j \in J_{train}}\right)$, where $\mathbb{Q}$ is the empirical quantile function. Trivial bins of zero size are removed. In subsection D.1, we demonstrate the usefulness of the proposed scheme on the synthetic GBDT-friendly dataset described in section 5.1 in Gorishniy et al. [13].

### 3.2.2 Building target-aware bins

In fact, there are also supervised approaches that employ training labels for constructing bins [11]. Intuitively, such target-aware algorithms aim to produce bins that correspond to relatively narrow ranges of possible target values. The supervised approach used in our work is identical in its spirit to the "C4.5 Discretization" algorithm from Kohavi and Sahami [23]. In a nutshell, for each feature, we recursively split its value range in a greedy manner using target as guidance, which is equivalent to building a decision tree (which uses for growing only this one feature and the target) and treating the regions corresponding to its leaves as the bins for PLE (see the illustration in Figure 4). Additionally, we define $b_0^i = \min_{j \in J_{train}} x_i^j$ and $b_T^i = \max_{j \in J_{train}} x_i^j$.

### 3.3 Periodic activation functions

Recall that in subsection 3.2 the work by Tancik et al. [42] was used as a starting point of our motivation for developing PLE. Thus, we also try to adapt the original work itself for tabular data problems. Our variation differs in two aspects. First, we take into account the fact the embedding framework described in subsection 3.1 forbids mixing features during the embedding process (see subsection D.2 for additional discussion). Second, we train the pre-activation coefficients instead of keeping them fixed. As a result, our approach is rather close to Li et al. [25] with the number of "groups" equal to the number of numerical features. We formalize the described scheme in Equation 2,

$$f_i(x) = \texttt{Periodic}(x) = \texttt{concat}[\sin(v), \cos(v)], \qquad v = [2\pi c_1 x, \ldots, 2\pi c_k x] \qquad (2)$$

where $c_i$ are trainable parameters initialized from $\mathcal{N}(0, \sigma)$. We observe that $\sigma$ is an important hyperparameter. Both $\sigma$ and $k$ are tuned using validation sets.

### 3.4 Simple differentiable layers

In the context of Deep Learning, embedding numerical features with conventional differentiable layers (e.g. linear layers, ReLU activation, etc.) is a natural approach. In fact, this technique is already used on its own in the recently proposed attention-based architectures [13, 24, 39] and in some models for CTR prediction problems [14, 40]. However, we also note that such conventional modules can be used on top of the components described in subsection 3.2 and subsection 3.3. In section 4, we find that such combinations often lead to better results.

## 4 Experiments

In this section, we empirically evaluate the techniques discussed in section 3 and compare them with Gradient Boosted Decision Trees to check the status quo of the "DL vs GBDT" competition.

### 4.1 Datasets

Table 1: Dataset properties. "RMSE" denotes root-mean-square error, "Acc." denotes accuracy.

|  | GE | CH | CA | HO | AD | OT | HI | FB | SA | CO | MI |
|---|---|---|---|---|---|---|---|---|---|---|---|
| #objects | 9873 | 10000 | 20640 | 22784 | 48842 | 61878 | 98049 | 197080 | 200000 | 581012 | 1200192 |
| #num. features | 32 | 10 | 8 | 16 | 6 | 93 | 28 | 50 | 200 | 54 | 136 |
| #cat. features | 0 | 1 | 0 | 0 | 8 | 0 | 0 | 1 | 0 | 0 | 0 |
| metric | Acc. | Acc. | RMSE | RMSE | Acc. | Acc. | Acc. | RMSE | Acc. | Acc. | RMSE |
| #classes | 5 | 2 | – | – | 2 | 9 | 2 | – | 2 | 7 | – |
| majority class | 29% | 79% | – | – | 76% | 26% | 52% | – | 89% | 48% | – |

We use eleven public datasets mostly from the previous works on tabular DL and Kaggle competitions. Importantly, we focus on the middle and large scale tasks, and our benchmark is biased towards GBDT-friendly problems, since, as of now, closing the gap with GBDT models on such tasks is one of the main challenges for tabular DL. The main dataset properties are summarized in Table 1 and the used sources and additional details are provided in Appendix C.

### 4.2 Implementation details

We mostly follow Gorishniy et al. [13] in terms of the hyperparameter tuning, training and evaluation protocols. Nevertheless, for completeness, we list all the details in Appendix E. In the next paragraph, we describe the implementation details specific to embeddings for numerical features.

**Embeddings for numerical features.** If linear layers are used, we tune their output dimensions. The PLE hyperparameters are the same for all features. For quantile-based PLE, we tune the number of quantiles. For target-aware PLE, we tune the following parameters for decision trees: the maximum number of leaves, the minimum number of items per leaf, and the minimum information gain required for making a split when growing the tree. For the Periodic module (see Equation 2), we tune $\sigma$ and $k$ (these hyperparameters are the same for all features).

## 4.3 Model names

In the experiments, we consider different combinations of backbones and embeddings. For convenience, we use the "Backbone-Embedding" pattern to name the models, where "Backbone" denotes the backbone (e.g. MLP, ResNet, Transformer) and "Embedding" denotes the embedding type. See Table 2 for all considered embedding modules. Note that:

- `Periodic` is defined in Equation 2.
- $PLE_q$ denotes the quantile-based PLE. $PLE_t$ denotes the target-aware PLE.
- `Linear_` denotes bias-free linear layer. LReLU denotes leaky ReLU. AutoDis was proposed in Guo et al. [14]
- "Transformer-L" is equivalent to FT-Transformer [13].

Table 2: Embedding names. See subsection 4.3

| Name | Embedding function ($f_i$) |
|---|---|
| L | Linear |
| LR | ReLU ∘ Linear |
| LRLR | ReLU ∘ Linear ∘ ReLU ∘ Linear |
| Q | $PLE_q$ |
| Q-L | Linear ∘ $PLE_q$ |
| Q-LR | ReLU ∘ Linear ∘ $PLE_q$ |
| Q-LRLR | ReLU ∘ Linear ∘ ReLU ∘ Linear ∘ $PLE_q$ |
| T | $PLE_t$ |
| T-L | Linear ∘ $PLE_t$ |
| T-LR | ReLU ∘ Linear ∘ $PLE_t$ |
| T-LRLR | ReLU ∘ Linear ∘ ReLU ∘ Linear ∘ $PLE_t$ |
| P | Periodic |
| PL | Linear ∘ Periodic |
| PLR | ReLU ∘ Linear ∘ Periodic |
| PLRLR | ReLU ∘ Linear ∘ ReLU ∘ Linear ∘ Periodic |
| AutoDis | Linear ∘ SoftMax ∘ Linear_ ∘ LReLU ∘ Linear_ |

## 4.4 Simple differentiable embedding modules

Table 3: Results for MLP equipped with simple embedding modules (see subsection 4.3). The metric values averaged over 15 random seeds are reported. The standard deviations are provided in Appendix F. We consider one result to be better than another if its mean score is better and its standard deviation is less than the difference. For each dataset, top results are in **bold**. Notation: ↓ corresponds to RMSE, ↑ corresponds to accuracy

| | GE↑ | CH↑ | CA↓ | HO↓ | AD↑ | OT↑ | HI↑ | FB↓ | SA↑ | CO↑ | MI↓ |
|---|---|---|---|---|---|---|---|---|---|---|---|
| MLP | **0.632** | 0.856 | 0.495 | 3.204 | 0.854 | 0.818 | 0.720 | 5.686 | 0.912 | **0.964** | 0.747 |
| MLP-L | **0.639** | **0.861** | 0.475 | 3.123 | **0.856** | **0.820** | 0.723 | 5.684 | 0.916 | 0.963 | 0.748 |
| MLP-LR | **0.642** | **0.860** | **0.471** | **3.084** | **0.857** | 0.819 | **0.726** | **5.625** | **0.923** | 0.963 | **0.746** |

We start by evaluating embedding modules consisting of "conventional" differentiable layers (linear layers, ReLU activations, etc.). The results are summarized in Table 3.
**The main takeaways**:

- first and foremost, the results indicate that MLP can benefit from embedding modules. Thus, we conclude that this backbone is worth attention when it comes to evaluating embedding modules.
- the simple LR module leads to modest, but consistent improvements when applied to MLP.

Interestingly, the "redundant" MLP-L configuration also tends to outperform the vanilla MLP. Although the improvements are not dramatic, the special property of this architecture is that the linear embedding module can be fused together with the first linear layer of MLP after training, which completely removes the overhead. As for LRLR and AutoDis, we observe that these heavy modules do not justify the extra costs (see the results in Appendix F).

## 4.5 Piecewise linear encoding

In this section, we evaluate the encoding scheme described in subsection 3.2. The results are summarized in Table 4.
**The main takeaways**:

- The piecewise linear encoding is often beneficial for both types of architectures (MLP and Transformer) and the profit can be significant (for example, see the CA and AD datasets).
- Adding differentiable components on top of the PLE can improve the performance. Though, the most expensive modifications such as Q-LRLR and T-LRLR are not worth it (see Appendix F).

Note that the benchmark is biased towards GBDT-friendly problems, so the typical superiority of tree-based bins over quantile-based bins, which can be observed in Table 4, may not generalize to more DL-friendly datasets. Thus, we do not make any general claims about the relative advantages of the two schemes here.

Table 4: Results for MLP and Transformer with embedding modules based on the piecewise linear encoding (subsection 3.2). Notation follows Table 3 and Table 2. The best results are defined separately for the MLP and Transformer backbones.

| | GE↑ | CH↑ | CA↓ | HO↓ | AD↑ | OT↑ | HI↑ | FB↓ | SA↑ | CO↑ | MI↓ |
|---|---|---|---|---|---|---|---|---|---|---|---|
| MLP | 0.632 | 0.856 | 0.495 | 3.204 | 0.854 | 0.818 | 0.720 | 5.686 | 0.912 | 0.964 | **0.747** |
| MLP-Q | **0.653** | 0.854 | 0.464 | **3.163** | 0.859 | 0.816 | 0.721 | 5.766 | 0.922 | 0.968 | 0.750 |
| MLP-T | **0.647** | **0.861** | 0.447 | **3.149** | 0.864 | **0.821** | 0.720 | 5.577 | 0.923 | 0.967 | 0.749 |
| MLP-Q-LR | **0.646** | 0.857 | 0.455 | 3.184 | 0.863 | 0.811 | 0.720 | **5.394** | 0.923 | **0.969** | 0.747 |
| MLP-T-LR | 0.640 | **0.861** | **0.439** | 3.207 | **0.868** | 0.818 | **0.724** | **5.508** | **0.924** | **0.968** | 0.747 |
| Transformer-L | 0.632 | 0.860 | 0.465 | 3.239 | 0.858 | **0.817** | 0.725 | **5.602** | 0.924 | 0.971 | **0.746** |
| Transformer-Q-L | **0.659** | 0.856 | 0.451 | 3.319 | 0.867 | 0.812 | **0.729** | 5.741 | **0.924** | **0.973** | 0.747 |
| Transformer-T-L | **0.663** | **0.861** | 0.454 | **3.197** | **0.871** | **0.817** | 0.726 | 5.803 | **0.924** | **0.974** | 0.747 |
| Transformer-Q-LR | **0.659** | 0.857 | 0.448 | 3.270 | 0.867 | 0.812 | 0.723 | 5.683 | 0.923 | 0.972 | 0.748 |
| Transformer-T-LR | **0.665** | 0.860 | **0.442** | **3.219** | 0.870 | **0.818** | **0.729** | 5.699 | **0.924** | 0.973 | 0.747 |

## 4.6 Periodic activation functions

Table 5: Results for MLP and Transformer with embedding modules based on periodic activations (subsection 3.3). Notation follows Table 3 and Table 2. The best results are defined separately for the MLP and Transformer backbones.

| | GE↑ | CH↑ | CA↓ | HO↓ | AD↑ | OT↑ | HI↑ | FB↓ | SA↑ | CO↑ | MI↓ |
|---|---|---|---|---|---|---|---|---|---|---|---|
| MLP | 0.632 | 0.856 | 0.495 | 3.204 | 0.854 | **0.818** | 0.720 | 5.686 | 0.912 | 0.964 | 0.747 |
| MLP-P | 0.631 | **0.860** | 0.489 | 3.129 | 0.869 | 0.807 | 0.723 | 5.845 | 0.923 | 0.968 | 0.747 |
| MLP-PL | 0.641 | **0.859** | **0.467** | 3.113 | 0.868 | **0.819** | 0.727 | **5.530** | 0.924 | **0.969** | 0.746 |
| MLP-PLR | **0.674** | 0.857 | **0.467** | **3.050** | 0.870 | 0.819 | 0.728 | **5.525** | 0.924 | 0.970 | 0.746 |
| Transformer-L | **0.632** | 0.860 | **0.465** | 3.239 | 0.858 | **0.817** | 0.725 | **5.602** | 0.924 | 0.971 | 0.746 |
| Transformer-PLR | 0.646 | 0.863 | 0.464 | 3.162 | 0.870 | 0.814 | **0.730** | 5.760 | 0.924 | 0.972 | 0.746 |

In this section, we evaluate embedding modules based on periodic activation functions as described in subsection 3.3. The results are reported in Table 5.

**The main takeaway**: on average, MLP-P is superior to the vanilla MLP. However, adding a differentiable component on top of the `Periodic` module should be the default strategy (which is in line with Li et al. [25]). Indeed, MLP-PLR and MLP-PL provide meaningful improvements over MLP-P (e.g. see GE, CA, HO) and even "fix" MLP-P where it is inferior to MLP (OT, FB).

Although MLP-PLR is usually superior to MLP-PL, we note that in the latter case the last linear layer of the embedding module is "redundant" in terms of expressivity and can be fused with the first linear layer of the backbone after training, which, in theory, can lead to a more lightweight model. Finally, we observe that MLP-PLRLR and MLP-PLR do not differ significantly enough to justify the extra cost of the `PLRLR` module (see Appendix F).

## 4.7 Comparing DL models and GBDT

In this section, we perform a big comparison of different approaches to identify the best embedding modules and backbones, as well as to check if embeddings for numerical features allow DL models to compete with GBDT on more tasks than before. Importantly, we compare *ensembles* of DL models against *ensembles* of GBDT, since Gradient Boosting is essentially an ensembling technique, so such comparison will be fairer. Note that we focus only on the best metric values without taking efficiency into account, so we only check if DL models are conceptually ready to compete with GBDT.

We consider three backbones: MLP, ResNet, and Transformer, since they are reported to be representative of what baseline DL backbones are currently capable of [13, 18, 24, 39]. Note that we do not include the attention-based models that also apply attention on the level of *objects* [24, 35, 39], since this non-parametric component is orthogonal to the central topic of our work. The results are summarized in Table 6.

Table 6: Results for ensembles of GBDT, the baseline DL models and their modifications using different types of embeddings for numerical features. Notation follows Table 3 and Table 2. Due to the limited precision, some *different* values are represented with the same figures.

| | GE ↑ | CH ↑ | CA ↓ | HO ↓ | AD ↑ | OT ↑ | HI ↑ | FB ↓ | SA ↑ | CO ↑ | MI ↓ | Avg. Rank |
|---|---|---|---|---|---|---|---|---|---|---|---|---|
| CatBoost | 0.692 | 0.861 | 0.430 | 3.093 | 0.873 | 0.825 | 0.727 | 5.226 | 0.924 | 0.967 | **0.741** | 3.6 ± 2.9 |
| XGBoost | 0.683 | 0.859 | 0.434 | 3.152 | **0.875** | 0.827 | 0.726 | 5.338 | 0.919 | 0.969 | 0.742 | 4.6 ± 2.7 |
| MLP | 0.665 | 0.856 | 0.486 | 3.109 | 0.856 | 0.822 | 0.727 | 5.616 | 0.913 | 0.968 | 0.746 | 8.5 ± 2.6 |
| MLP-LR | 0.679 | 0.861 | 0.463 | 3.012 | 0.859 | 0.826 | 0.731 | 5.477 | 0.924 | 0.972 | 0.744 | 5.5 ± 2.7 |
| MLP-Q-LR | 0.682 | 0.859 | 0.433 | 3.080 | 0.867 | 0.818 | 0.724 | **5.144** | 0.924 | 0.974 | 0.745 | 5.1 ± 1.9 |
| MLP-T-LR | 0.673 | 0.861 | 0.435 | 3.099 | 0.870 | 0.821 | 0.727 | 5.409 | 0.924 | 0.973 | 0.746 | 5.1 ± 1.7 |
| MLP-PLR | **0.700** | 0.858 | 0.453 | **2.975** | 0.874 | **0.830** | **0.734** | 5.388 | **0.924** | 0.975 | 0.743 | 3.0 ± 2.4 |
| ResNet | 0.690 | 0.861 | 0.483 | 3.081 | 0.856 | 0.821 | 0.734 | 5.482 | 0.918 | 0.968 | 0.745 | 6.7 ± 3.3 |
| ResNet-LR | 0.672 | 0.862 | 0.450 | 2.992 | 0.859 | 0.822 | 0.733 | 5.415 | 0.923 | 0.971 | 0.743 | 5.6 ± 2.7 |
| ResNet-Q-LR | 0.674 | 0.859 | 0.427 | 3.066 | 0.868 | 0.815 | 0.729 | 5.309 | 0.923 | 0.976 | 0.746 | 4.7 ± 2.0 |
| ResNet-T-LR | 0.683 | 0.862 | **0.425** | 3.030 | 0.872 | 0.822 | 0.731 | 5.471 | 0.923 | 0.975 | 0.744 | 4.1 ± 1.9 |
| ResNet-PLR | 0.691 | 0.861 | 0.443 | 3.040 | **0.874** | 0.825 | 0.734 | 5.400 | 0.924 | 0.975 | 0.743 | 3.2 ± 1.3 |
| Transformer-L | 0.668 | 0.861 | 0.455 | 3.188 | 0.860 | 0.824 | 0.727 | 5.434 | 0.924 | 0.973 | 0.743 | 5.9 ± 2.2 |
| Transformer-LR | 0.666 | 0.861 | 0.446 | 3.193 | 0.861 | 0.824 | 0.733 | 5.430 | 0.924 | 0.973 | 0.743 | 5.2 ± 2.2 |
| Transformer-Q-LR | 0.690 | 0.857 | **0.425** | 3.143 | 0.868 | 0.818 | 0.726 | 5.471 | **0.924** | 0.975 | 0.744 | 4.4 ± 2.2 |
| Transformer-T-LR | 0.686 | 0.862 | **0.423** | 3.149 | 0.871 | 0.823 | 0.733 | 5.515 | 0.924 | **0.976** | 0.744 | 3.7 ± 2.2 |
| Transformer-PLR | 0.686 | **0.864** | 0.449 | 3.091 | 0.873 | 0.823 | 0.734 | 5.581 | **0.924** | 0.975 | 0.743 | 3.9 ± 2.5 |

**The main takeaways for DL models**:

- For most datasets, embeddings for numerical features can provide noticeable improvements for three different backbones. Although the average rank is not a good metric for making subtle conclusions, we highlight the impressive difference in average ranks between the MLP and MLP-PLR models.

- The simplest LR embedding is a good baseline solution: although the performance gains are not dramatic, its main advantage is consistency (e.g. see MLP vs MLP-LR).

- The PLR module provides the best average performance. Empirically, we observe $\sigma$ (see Equation 2) to be an important hyperparameter that should be tuned.

- Piecewise linear encoding (PLE) allows building well performing embeddings (e.g. T-LR, Q-LR). In addition to that, PLE itself is worth attention because of its simplicity, interpretability and efficiency (no computationally expensive periodic functions).

- Importantly, after the MLP-like architectures are coupled with embeddings for numerical features, they perform on par with the Transformer-based models.

**The main takeaway for the "DL vs GBDT" competition**: embeddings for numerical features is a significant design aspect that has a great potential for improving DL models and closing the gap with GBDT on GBDT-friendly tasks. Let us illustrate this claim with several observations:

- The benchmark is initially biased to GBDT-friendly problems, which can be observed by comparing GBDT solutions with the vanilla DL models (MLP, ResNet, Transformer-L).

- However, for the vast majority of the "backbone & dataset" pairs, proper embeddings are the only thing needed to close the gap with GBDT. Exceptions (rather formal) include the MI dataset and the following pairs: "ResNet & GE", "Transformer & FB", "Transformer & GE", "Transformer & OT".

- Additionally, to the best of our knowledge, it is the first time when DL models perform on par with GBDT on the well-known California Housing and Adult datasets.

That said, compared to GBDT models, efficiency can still be an issue for the considered DL architectures. In any case, the trade-off completely depends on the specific use case and requirements.

# 5 Analysis

## 5.1 Comparing model sizes

To quantify the effect of embeddings for numerical features on model sizes, we report the parameter counts in Table 7. Overall, introducing embeddings for numerical features can cause non-negligible overhead in terms of model size. Importantly, the overhead in terms of size does not translate to the same overhead in terms of training times and throughput. For example, the almost 2000-fold increase in the parameter count for MLP-LR on the CH dataset results in only 1.5-fold increase in training times. Finally, in practice, we observe that coupling MLP and ResNet with embedding modules leads to architectures that are still faster than Transformer-based models.

Table 7: Parameter counts for MLP with different embedding modules. All the models are tuned and the corresponding backbones are not identical in their sizes, so we take into account the fact that different approaches require a different number of parameters to realize their full potential.

|  | GE | CH | CA | HO | AD | OT | HI | FB | SA | CO | MI |
|---|---|---|---|---|---|---|---|---|---|---|---|
| MLP | 2.0M | 1.5K | 43.5K | 3.6M | 5.3M | 479.9K | 25.8K | 937.3K | 5.8M | 3.2M | 276.5K |
| MLP-LR | ×2.52 | ×1931.03 | ×25.05 | ×1.28 | ×0.35 | ×12.53 | ×68.16 | ×4.76 | ×1.58 | ×0.72 | ×15.79 |
| MLP-T | ×1.58 | ×14.13 | ×7.97 | ×0.43 | ×0.04 | ×2.27 | ×5.85 | ×0.47 | ×0.59 | ×0.74 | ×3.85 |
| MLP-T-LR | ×1.61 | ×463.55 | ×6.80 | ×0.23 | ×0.16 | ×2.52 | ×113.22 | ×3.43 | ×0.41 | ×0.35 | ×8.47 |
| MLP-PLR | ×1.73 | ×250.24 | ×12.94 | ×1.07 | ×0.66 | ×8.05 | ×110.57 | ×4.93 | ×0.64 | ×0.44 | ×9.57 |

## 5.2 Ablation study

Table 8: Comparing piecewise linear encoding (PLE) with the two variations described in subsection 5.2. Notation follows Table 3 and Table 2.

|  | GE ↑ | CH ↑ | CA ↓ | HO ↓ | AD ↑ | OT ↑ | HI ↑ | FB ↓ |
|---|---|---|---|---|---|---|---|---|
| MLP-Q (piecewise linear) | **0.653** | **0.854** | 0.464 | **3.163** | 0.859 | **0.816** | **0.721** | 5.766 |
| MLP-Q (binary) | **0.652** | 0.815 | 0.462 | 3.200 | **0.860** | 0.810 | **0.720** | 5.748 |
| MLP-Q (one-blob) | 0.613 | 0.851 | **0.461** | 3.187 | 0.857 | 0.808 | 0.719 | **5.645** |
| MLP-T (piecewise linear) | **0.647** | **0.861** | **0.447** | 3.149 | 0.864 | **0.821** | 0.720 | 5.577 |
| MLP-T (binary) | 0.639 | 0.855 | 0.464 | **3.163** | 0.869 | 0.813 | 0.718 | 5.572 |
| MLP-T (one-blob) | 0.622 | 0.858 | 0.464 | **3.158** | **0.870** | 0.809 | **0.724** | **5.475** |

In this section, we compare two alternative binning-based encoding schemes with PLE (see subsection 3.2). The first one ("thermometer" [6]) sets the value 1 instead of the piecewise linear term (see Equation 1). The second one is a generalized version of the one-blob encoding [29] (see subsection E.1 for details). The tuning and evaluation protocols are the same as in subsection 4.2. The results in table Table 8 indicate that making the binning-based encoding piecewise linear is a good default strategy.

## 5.3 Piecewise linear encoding as a feature preprocessing technique

It is known that data preprocessing, such as standardization or quantile transformation, is often crucial for DL models for achieving competitive performance. Moreover, the performance can significantly vary between different types of preprocessing. At the same time, PLE-representations contain only values from $[0, 1]$ and they are invariant to shifting and scaling, which makes PLE itself a general feature preprocessing technique potentially suitable for DL models without the need to use traditional preprocessing first.

To illustrate that, for datasets where the quantile transformation was used in section 4, we reevaluate the tuned configurations of MLP, MLP-Q, and MLP-T with different preprocessing policies and report the results in Table 9 (note that standardization is equivalent to no preprocessing for models with PLE).

Table 9: Results for MLP and MLP with `PLE` for different types of data preprocessing. Solutions using PLE are significantly less sensitive to data preprocessing. Notation follows Table 3 and Table 2.

| | GE ↑ | CH ↑ | CA ↓ | HO ↓ | AD ↑ | HI ↑ | FB ↓ | SA ↑ | CO ↑ | MI ↓ |
|---|---|---|---|---|---|---|---|---|---|---|
| MLP (none) | 0.565 | 0.796 | 1.118 | 5.328 | 0.808 | 0.707 | 13.125 | 0.911 | 0.948 | 0.844 |
| MLP (standard) | 0.629 | 0.855 | 0.509 | 3.303 | 0.855 | 0.721 | 5.919 | 0.912 | 0.963 | 0.754 |
| MLP (quantile) | 0.632 | 0.856 | 0.495 | 3.204 | 0.854 | 0.720 | 5.686 | 0.912 | 0.964 | 0.747 |
| MLP-Q (none) | 0.654 | 0.851 | 0.463 | 3.162 | 0.860 | 0.721 | 5.889 | 0.922 | 0.968 | 0.754 |
| MLP-Q (quantile) | 0.653 | 0.854 | 0.464 | 3.163 | 0.859 | 0.721 | 5.766 | 0.922 | 0.968 | 0.750 |
| MLP-T (none) | 0.644 | 0.860 | 0.447 | 3.175 | 0.865 | 0.721 | 5.598 | 0.923 | 0.968 | 0.749 |
| MLP-T (quantile) | 0.647 | 0.861 | 0.447 | 3.149 | 0.864 | 0.720 | 5.577 | 0.923 | 0.967 | 0.749 |

First, the vanilla MLP often becomes unusable without preprocessing. Second, for the vanilla MLP, it can be important to choose one specific type of preprocessing (CA, HO, FB, MI), which is less pronounced for MLP-Q and not the case for MLP-T (though, this specific observation can be the property of the benchmarks, not of MLP-T). Overall, the results indicate that models using `PLE` are less sensitive to the initial preprocessing compared to the vanilla MLP. This is an additional benefit of PLE-representations for practitioners since the aspect of preprocessing becomes less critical with PLE.

### 5.4 The "feature engineering" perspective

Table 10: The comparison of the effects of `Periodic`-based modules for XGBoost and MLP

| | CA ↓ | HO ↓ | HI ↑ |
|---|---|---|---|
| XGBoost | 0.436 | 3.160 | 0.724 |
| XGBoost with `Periodic` | 0.441 | 3.184 | 0.724 |
| MLP | 0.495 | 3.204 | 0.720 |
| MLP-PL | 0.467 | 3.113 | 0.727 |

At first sight, feature embeddings may resemble feature engineering and should be suitable for all kinds of models. However, the proposed embedding schemes are motivated by DL-specific aspects of training (see the motivational parts of subsection 3.2 and subsection 3.3). While our methods are likely to transfer well to models with similar training properties (e.g. to linear models since those are a special case of deep models), it is not the case in general. To illustrate that, we try adopting the `Periodic` module for XGBoost by fixing the random coefficients from Equation 2. We also keep the original features instead of dropping them. The tuning and evaluation protocols are the same as in subsection 4.2. The results in Table 10 show that this technique, while being useful for DL models, does not provide any benefits for XGBoost.

## 6 Conclusion & Future work

In this work, we have demonstrated that embeddings for numerical features are an important design aspect of tabular DL architectures. Namely, it allows existing DL backbones to achieve noticeably better results and significantly reduce the gap with Gradient Boosted Decision Trees. We have described two approaches illustrating this phenomenon, one using the piecewise linear encoding of original scalar values, and another using periodic functions. We have also shown that traditional MLP-like models coupled with embeddings can perform on par with attention-based models.

Nevertheless, we have only scratched the surface of the new direction. For example, it is still to be explained how exactly the discussed embedding modules help optimization on the fundamental level. Additionally, we have considered only schemes where the same functional transformation was applied to all features, which may be a suboptimal choice.

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
