# Supplementary material

## A    MLP with embeddings for numerical features

We provide visual explanation of how embeddings are passed to MLP in Figure 2 and Figure 3. Also, we provide the formal explanation in Equation 3 (categorical features are omitted for simplicity).

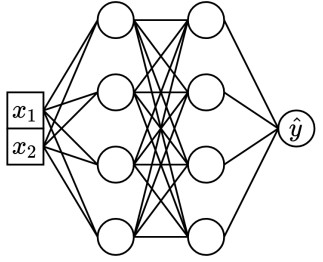

Figure 2: The vanilla MLP. The model takes two numerical features as input.

Figure 3: The same MLP as in Figure 2, but now with embeddings for numerical features.

$$\text{MLP}(z_1, \ldots, z_k) = \text{MLP}\left(\text{concat}[z_1, \ldots, z_k]\right) \qquad \text{concat}[z_1, \ldots, z_k] \in \mathbb{R}^{d_1 + \cdots + d_k} \qquad (3)$$

## B    Target-aware piecewise linear encoding

We provide visualisation of target-aware PLE (subsubsection 3.2.2) in Figure 4.

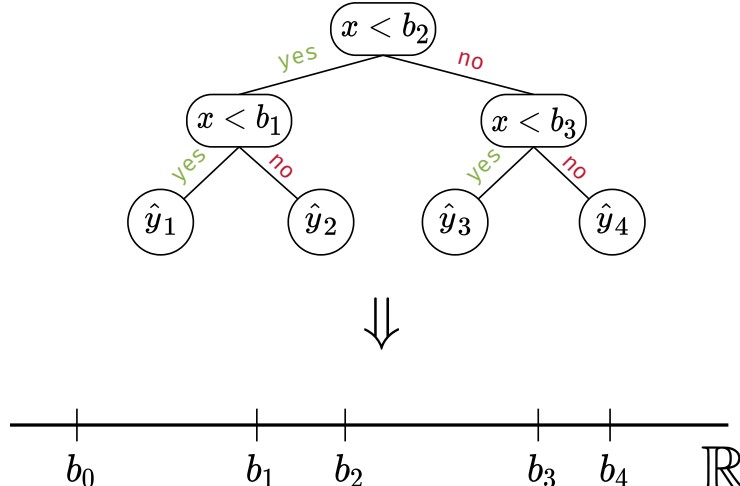

Figure 4: Obtaining bins for PLE from decision trees.

# C  Additional details on datasets

Table 11: Details on datasets, used for experiments

| Abbr | Name | # Train | # Validation | # Test | # Num | # Cat | Task type | Batch size |
|------|------|---------|--------------|--------|-------|-------|-----------|------------|
| GE | Gesture Phase | 6318 | 1580 | 1975 | 32 | 0 | Multiclass | 128 |
| CH | Churn Modelling | 6400 | 1600 | 2000 | 10 | 1 | Binclass | 128 |
| CA | California Housing | 13209 | 3303 | 4128 | 8 | 0 | Regression | 256 |
| HO | House 16H | 14581 | 3646 | 4557 | 16 | 0 | Regression | 256 |
| AD | Adult | 26048 | 6513 | 16281 | 6 | 8 | Binclass | 256 |
| OT | Otto Group Products | 39601 | 9901 | 12376 | 93 | 0 | Multiclass | 512 |
| HI | Higgs Small | 62751 | 15688 | 19610 | 28 | 0 | Binclass | 512 |
| FB | Facebook Comments Volume | 157638 | 19722 | 19720 | 50 | 1 | Regression | 512 |
| SA | Santander Customer Transactions | 128000 | 32000 | 40000 | 200 | 0 | Binclass | 1024 |
| CO | Covertype | 371847 | 92962 | 116203 | 54 | 0 | Multiclass | 1024 |
| MI | MSLR-WEB10K (Fold 1) | 723412 | 235259 | 241521 | 136 | 0 | Regression | 1024 |

We used the following datasets:

- Gesture Phase Prediction (Madeo et al. [27])
- Churn Modeling[2]
- California Housing (real estate data, Kelley Pace and Barry [20])
- House 16H[3]
- Adult (income estimation, Kohavi [22])
- Otto Group Product Classification[4]
- Higgs (simulated physical particles, Baldi et al. [4]; we use the version with 98K samples available in the OpenML repository [44])
- Santander Customer Transaction Prediction[5]
- Facebook Comments (Singh et al. [37])
- Covertype (forest characteristics, Blackard and Dean. [5])
- Microsoft (search queries, Qin and Liu [33]). We follow the pointwise approach to learning-to-rank and treat this ranking problem as a regression problem.

# D  Additional analysis

## D.1  Testing quantile-based `PLE` on the synthetic GBDT-friendly dataset

In this section, we apply the quantile-based piecewise linear encoding (described in subsubsection 3.2.1 to MLP and Transformer on the synthetic GBDT-friendly dataset described in section 5.1 in Gorishniy et al. [13]. In a nutshell, features of this dataset are sampled randomly from $\mathcal{N}(0, 1)$, and the target is produced by an ensemble of randomly constructed decision trees applied to the sampled features. This task turns out to be easy for GBDT, but hard for traditional DL models [13]. The results are visualized in Figure 5. As the plot shows, `PLE`-representations can be helpful for both MLP and Transformer backbones. In the considered synthetic setup, increasing the number of bins leads to better results, however, in practice, using too many bins can lead to overfitting; therefore, we recommend tuning the number of bins based on a validation set.

**Technical details.** Our dataset has $10,000$ objects, $8$ features and the target was produced by $16$ decision trees of depth $6$. CatBoost is trained with the default hyperparameters. Transformer is trained with the default hyperparameters of FT-Transformer. The MLP backbone has four layers of size $256$ each. Importantly, the task GBDT-friendly, which can be illustrated by the performance of the *tuned* MLP: $0.2229 \pm 0.0055$ (it is still worse than the performance of CatBoost). The remaining details can be found in the source code.

---

[2]https://www.kaggle.com/shrutimechlearn/churn-modelling

[3]https://www.openml.org/d/574

[4]https://www.kaggle.com/c/otto-group-product-classification-challenge/data

[5]https://www.kaggle.com/c/santander-customer-transaction-prediction

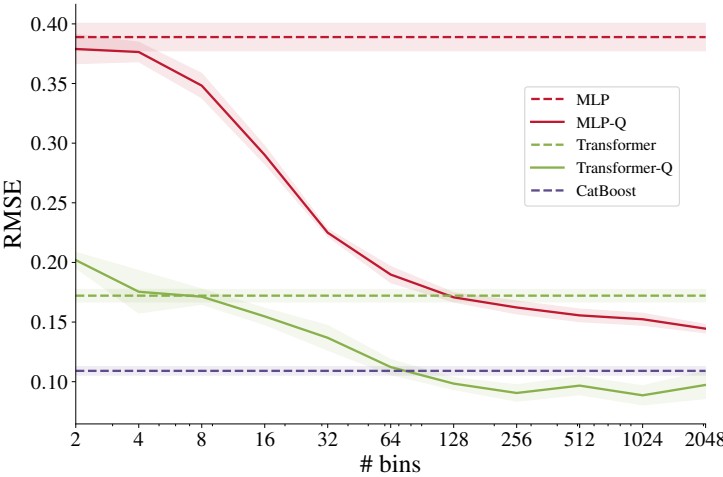

Figure 5: RMSE (averaged over five random seeds) of different approaches on the same synthetic GBDT-friendly task. Using PLE-representations ("-Q") instead of scalar values improves the performance of MLP and Transformer. Note that in practice, increasing the number of bins does not always lead to better results.

## D.2 Fourier features

In this section, we test Fourier features implemented exactly as in Tancik et al. [42], i.e. pre-activation coefficients are not trained and features are mixed right from the start. Importantly, the latter means that this approach is not covered by the embedding framework described in subsection 3.1. As reported in Table 12, MLP equipped with the original Fourier features does not perform well even compared to the vanilla MLP. So, it seems to be important to embed each feature separately as described in subsection 3.1.

Table 12: Results for the vanilla MLP and MLP equipped with Fourier features [42]. Notation follows Table 3 and Table 2.

|  | GE ↑ | CH ↑ | CA ↓ | HO ↓ | AD ↑ | OT ↑ | HI ↑ | FB ↓ | SA ↑ | CO ↑ | MI ↓ |
|---|---|---|---|---|---|---|---|---|---|---|---|
| MLP | **0.632** | **0.856** | **0.495** | **3.204** | 0.854 | **0.818** | **0.720** | **5.686** | 0.912 | **0.964** | **0.747** |
| MLP (Fourier features) | 0.612 | 0.845 | **0.495** | 3.267 | **0.858** | 0.810 | 0.711 | 5.767 | **0.915** | 0.961 | 0.749 |

## E  Implementation details

We mostly follow Gorishniy et al. [13] in terms of the tuning, training and evaluation protocols.

**Data preprocessing**. Preliminary data preprocessing is known to be crucial for the optimization of tabular DL models. For each dataset, the same preprocessing was used for all deep models for a fair comparison. For all datasets except for Otto Group Product Classification, we use the quantile transformation from the Scikit-learn library [30]. For Otto Group Product Classification, we do not apply any feature preprocessing. We also apply standardization to regression targets for all algorithms.

**Tuning**. For every dataset, we carefully tune each model's hyperparameters. The best hyperparameters are the ones that perform best on the validation set, so the test set is never used for tuning. For most algorithms, we use the Optuna library [1] to run Bayesian optimization (the Tree-Structured Parzen Estimator algorithm), which is reported to be superior to random search [43]. The search spaces for all hyperparameters are reported in the appendix.

**Evaluation**. For each tuned configuration, we run 15 experiments with different random seeds and report the average performance on the test set.

**Ensembles**. For each model-dataset pair, we obtain three ensembles by splitting the 15 single models into three disjoint groups of equal size and averaging predictions of single models within each group.

**Neural networks**. The implementations of the MLP, ResNet, and Transformer backbones are taken from Gorishniy et al. [13]. We minimize cross-entropy for classification problems and mean squared error for regression problems. We use the AdamW optimizer [26]. We do not apply learning rate schedules. For each dataset, we use a predefined batch size (see Appendix C for the specific values). We continue training until there are `patience` $+ 1$ consecutive epochs without improvements on the validation set; we set `patience` $= 16$ for all models.

**Categorical features.** For CatBoost, we employ the built-in support for categorical features. For all other algorithms, we use the one-hot encoding.

## E.1 One-blob encoding

In subsection 5.2, we used a slightly generalized version of the original one-blob encoding [29]. Namely, while the original sets the width of the kernel to $T^{-1}$ ($T$ is the number of bins), we set it to $T^{-\gamma}$ and tune $\gamma$.

## E.2 Hyperparameter tuning configurations

## E.3 CatBoost

We fix and do not tune the following hyperparameters:

- `early-stopping-rounds` $= 50$
- `od-pval` $= 0.001$
- `iterations` $= 2000$

For tuning on the MI and CO datasets, we set the `task_type` parameter to "GPU". In all other cases (including the evaluation on these two datasets), we set this parameter to "CPU".

Table 13: CatBoost hyperparameter space

| Parameter | Distribution |
|---|---|
| Max depth | UniformInt$[1, 10]$ |
| Learning rate | LogUniform$[0.001, 1]$ |
| Bagging temperature | Uniform$[0, 1]$ |
| L2 leaf reg | LogUniform$[1, 10]$ |
| Leaf estimation iterations | UniformInt$[1, 10]$ |
| # Iterations | 100 |

## E.4 XGBoost

We fix and do not tune the following hyperparameters:

- `booster` $=$ "gbtree"
- `early-stopping-rounds` $= 50$
- `n-estimators` $= 2000$

Table 14: XGBoost hyperparameter space.

| Parameter | Distribution |
|---|---|
| Max depth | UniformInt[3, 10] |
| Min child weight | LogUniform[0.0001, 100] |
| Subsample | Uniform[0.5, 1] |
| Learning rate | LogUniform[0.001, 1] |
| Col sample by tree | Uniform[0.5, 1] |
| Gamma | {0, LogUniform[0.001, 100]} |
| Lambda | {0, LogUniform[0.1, 10]} |
| # Iterations | 100 |

## E.5   MLP

Table 15: MLP hyperparameter space.

| Parameter | Distribution |
|---|---|
| # Layers | UniformInt[1, 16] |
| Layer size | UniformInt[1, 1024] |
| Dropout | {0, Uniform[0, 0.5]} |
| Learning rate | LogUniform[5$e$-5, 0.005] |
| Weight decay | {0, LogUniform[1$e$-6, 1$e$-3]} |
| # Iterations | 100 |

## E.6   ResNet

Table 16: ResNet hyperparameter space.

| Parameter | Distribution |
|---|---|
| # Layers | UniformInt[1, 8] |
| Layer size | UniformInt[32, 512] |
| Hidden factor | Uniform[1, 4] |
| Hidden dropout | Uniform[0, 0.5] |
| Residual dropout | {0, Uniform[0, 0.5]} |
| Learning rate | LogUniform[5$e$-5, 0.005] |
| Weight decay | {0, LogUniform[1$e$-6, 1$e$-3]} |
| # Iterations | 100 |

### E.7 Transformer

Table 17: Transformer hyperparameter space. Here (A) = {SA, CO, MI} and (B) = the rest

| Parameter | (Datasets) Distribution |
|---|---|
| # Layers | (A) $\mathrm{UniformInt}[2, 4]$, (B) $\mathrm{UniformInt}[1, 4]$ |
| Embedding size | (A) $\mathrm{UniformInt}[192, 512]$, (B) $\mathrm{UniformInt}[96, 512]$ |
| Residual dropout | (A) $\mathrm{Const}(0.0)$, (B) $\{0, \mathrm{Uniform}[0, 0.2]\}$ |
| Attention dropout | (A,B) $\mathrm{Uniform}[0, 0.5]$ |
| FFN dropout | (A,B) $\mathrm{Uniform}[0, 0.5]$ |
| FFN factor | (A,B) $\mathrm{Uniform}[2/3, 8/3]$ |
| Learning rate | (A) $\mathrm{LogUniform}[1e\text{-}5, 3e\text{-}4]$, (B) $\mathrm{LogUniform}[1e\text{-}5, 1e\text{-}3]$ |
| Weight decay | (A) $\mathrm{Const}(1e\text{-}5)$, (B) $\mathrm{LogUniform}[1e\text{-}6, 1e\text{-}4]$ |
| # Iterations | (A) 50, (B) 100 |

### E.8 Embedding hyperparameters

The distribution for the output dimensions of linear layers is $\mathrm{UniformInt}[1, 128]$.

PLE. We share the same hyperparameter space for PLE across all datasets and models. For the quantile-based PLE, the distribution for the number of quantiles is $\mathrm{UniformInt}[2, 256]$. For the target-aware (tree-based) PLE, the distribution for the number of leaves is $\mathrm{UniformInt}[2, 256]$, the distribution for the minimum number of items per leaf is $\mathrm{UniformInt}[1, 128]$ and the distribution for the minimum information gain required for making a split is $\mathrm{LogUniform}[1e\text{-}9, 0.01]$.

Periodic. The distribution for $k$ (see Equation 2) is $\mathrm{UniformInt}[1, 128]$.

## F Extended tables with experimental results

The scores with standard deviations for single models and ensembles are provided in Table 18 and Table 19 respectively. Please, refer to Table 2 to learn about the model names.

Additionally, we include the results for the DICE embeddings [41], which is a general way to represent numbers with vectors introduced in the context of NLP. The results though demonstrate that it is a suboptimal approach in tabular data problems.

Table 18: Extended results for single models

| | GE↑ | CH↑ | CA↓ | HO↓ | AD↑ | OT↑ | HI↑ | FB↓ | SA↑ | CO↑ | MI↓ |
|---|---|---|---|---|---|---|---|---|---|---|---|
| CatBoost | 0.683±4.7e-3 | 0.861±3.5e-3 | 0.433±1.8e-3 | 3.115±1.9e-2 | 0.872±9.0e-4 | 0.824±1.1e-3 | 0.726±1.0e-3 | 5.324±4.1e-2 | 0.923±3.6e-4 | 0.966±3.3e-4 | 0.743±3.1e-4 |
| XGBoost | 0.678±4.9e-3 | 0.858±2.2e-3 | 0.436±2.5e-3 | 3.160±6.9e-3 | 0.874±8.2e-4 | 0.825±2.3e-3 | 0.724±1.0e-3 | 5.383±2.9e-2 | 0.918±5.0e-4 | 0.969±6.1e-4 | 0.742±1.6e-4 |
| MLP | 0.632±1.4e-2 | 0.856±2.8e-3 | 0.495±4.3e-3 | 3.204±4.0e-2 | 0.854±1.6e-3 | 0.818±3.1e-3 | 0.720±2.3e-3 | 5.686±4.7e-2 | 0.912±4.3e-4 | 0.964±8.6e-4 | 0.747±2.5e-4 |
| MLP-L | 0.639±1.3e-2 | 0.861±2.1e-3 | 0.475±5.4e-3 | 3.123±4.5e-2 | 0.856±1.6e-3 | 0.820±1.5e-3 | 0.723±1.6e-3 | 5.684±4.5e-2 | 0.916±3.5e-4 | 0.963±9.3e-4 | 0.748±4.1e-4 |
| MLP-LR | 0.642±1.5e-2 | 0.860±3.0e-3 | 0.471±2.6e-3 | 3.084±2.7e-2 | 0.857±1.9e-3 | 0.819±1.6e-3 | 0.726±1.9e-3 | 5.625±5.6e-2 | 0.923±3.1e-4 | 0.963±1.4e-3 | 0.746±3.9e-4 |
| MLP-LRLR | 0.654±1.7e-2 | 0.861±2.6e-3 | 0.460±3.8e-3 | 3.070±3.6e-2 | 0.857±1.4e-3 | 0.819±2.2e-3 | 0.725±1.2e-3 | 5.551±4.6e-2 | 0.923±3.0e-4 | 0.963±1.4e-3 | 0.746±3.0e-4 |
| MLP-Q | 0.653±8.9e-3 | 0.854±3.0e-3 | 0.464±3.1e-3 | 3.163±3.1e-2 | 0.859±1.6e-3 | 0.816±2.6e-3 | 0.721±1.0e-3 | 5.766±5.3e-2 | 0.922±2.6e-4 | 0.968±6.9e-4 | 0.750±3.7e-4 |
| MLP-Q-LR | 0.646±6.3e-3 | 0.857±2.6e-3 | 0.455±3.4e-3 | 3.184±3.1e-2 | 0.863±1.7e-3 | 0.811±1.8e-3 | 0.720±1.5e-3 | 5.394±1.5e-1 | 0.923±6.1e-4 | 0.969±4.8e-4 | 0.747±3.9e-4 |
| MLP-Q-LRLR | 0.644±6.2e-3 | 0.859±2.2e-3 | 0.452±4.3e-3 | 3.118±4.6e-2 | 0.869±1.5e-3 | 0.812±2.5e-3 | 0.724±1.3e-3 | 5.618±2.0e-1 | 0.924±4.5e-4 | 0.969±1.2e-3 | 0.748±4.6e-4 |
| MLP-T | 0.647±5.7e-3 | 0.861±1.1e-3 | 0.447±2.0e-3 | 3.149±5.2e-2 | 0.864±6.3e-4 | 0.821±1.8e-3 | 0.720±1.9e-3 | 5.577±3.7e-2 | 0.923±3.0e-4 | 0.967±1.1e-3 | 0.749±4.4e-4 |
| MLP-T-LR | 0.640±6.9e-3 | 0.857±2.0e-3 | 0.439±3.7e-3 | 3.207±5.2e-2 | 0.868±1.1e-3 | 0.818±1.3e-3 | 0.724±1.7e-3 | 5.508±3.0e-2 | 0.924±2.4e-4 | 0.968±7.2e-4 | 0.747±5.7e-4 |
| MLP-T-LRLR | 0.629±1.0e-2 | 0.857±2.4e-3 | 0.446±3.6e-3 | 3.153±4.0e-2 | 0.870±9.9e-4 | 0.818±2.1e-3 | 0.725±1.3e-3 | 5.553±2.4e-2 | 0.924±3.6e-4 | 0.967±8.5e-4 | 0.748±5.8e-4 |
| MLP-P | 0.631±1.7e-2 | 0.860±3.1e-3 | 0.489±2.4e-3 | 3.129±4.3e-2 | 0.869±1.5e-3 | 0.807±4.3e-3 | 0.723±1.5e-3 | 5.845±6.4e-2 | 0.923±4.3e-4 | 0.968±9.0e-4 | 0.747±3.1e-4 |
| MLP-PL | 0.641±1.0e-2 | 0.859±2.4e-3 | 0.467±2.9e-3 | 3.113±3.1e-2 | 0.868±1.1e-3 | 0.819±1.7e-3 | 0.727±1.7e-3 | 5.530±9.5e-2 | 0.924±4.0e-4 | 0.969±5.0e-4 | 0.746±2.6e-4 |
| MLP-PLR | 0.674±1.0e-2 | 0.857±2.4e-3 | 0.467±5.8e-3 | 3.050±3.4e-2 | 0.870±1.0e-3 | 0.819±2.0e-3 | 0.728±1.6e-3 | 5.525±3.5e-2 | 0.924±4.0e-4 | 0.970±9.5e-4 | 0.746±3.0e-4 |
| MLP-PLRLR | 0.676±1.6e-2 | 0.863±3.1e-3 | 0.456±3.7e-3 | 3.038±2.3e-2 | 0.871±1.4e-3 | 0.818±1.7e-3 | 0.725±1.6e-3 | 5.606±8.9e-2 | 0.924±2.8e-4 | 0.968±2.0e-3 | 0.744±2.8e-4 |
| MLP-AutoDis | 0.649±1.2e-2 | 0.857±3.2e-3 | 0.474±5.1e-3 | 3.165±1.8e-2 | 0.859±1.3e-3 | 0.807±2.4e-3 | 0.725±1.9e-3 | 5.670±6.1e-2 | 0.924±3.0e-4 | 0.963±8.7e-4 | — |
| MLP-DICE | 0.610±1.2e-2 | 0.858±2.9e-3 | 0.491±3.0e-3 | 3.146±3.5e-2 | 0.860±1.4e-3 | 0.778±4.9e-3 | 0.720±9.8e-4 | 5.726±3.6e-2 | 0.920±4.8e-4 | 0.964±1.1e-3 | 0.748±2.9e-4 |
| ResNet | 0.655±2.0e-2 | 0.858±3.1e-3 | 0.490±5.0e-3 | 3.153±3.6e-2 | 0.855±8.9e-4 | 0.817±3.4e-3 | 0.729±2.1e-3 | 5.681±5.3e-2 | 0.916±5.0e-4 | 0.965±8.3e-4 | 0.747±4.1e-4 |
| ResNet-L | 0.644±1.9e-2 | 0.859±1.8e-3 | 0.490±6.6e-3 | 3.126±5.6e-2 | 0.855±1.4e-3 | 0.813±2.2e-3 | 0.730±9.7e-4 | 5.758±8.0e-2 | 0.915±4.3e-4 | 0.964±1.8e-3 | 0.747±4.7e-4 |
| ResNet-LR | 0.635±2.3e-2 | 0.861±2.2e-3 | 0.465±3.5e-3 | 3.096±5.8e-2 | 0.856±1.6e-3 | 0.815±1.3e-3 | 0.729±1.3e-3 | 5.574±7.4e-2 | 0.922±4.4e-4 | 0.967±8.8e-4 | 0.746±4.4e-4 |
| ResNet-Q | 0.658±8.0e-3 | 0.858±2.4e-3 | 0.454±3.6e-3 | 3.251±3.8e-2 | 0.860±1.3e-3 | 0.811±1.6e-3 | 0.718±1.0e-3 | 5.828±9.3e-2 | 0.921±9.1e-4 | 0.970±5.7e-4 | 0.749±2.9e-4 |
| ResNet-Q-LR | 0.650±9.2e-3 | 0.854±4.2e-3 | 0.446±5.1e-3 | 3.217±5.2e-2 | 0.865±2.2e-3 | 0.808±2.6e-3 | 0.722±1.9e-3 | 5.514±6.0e-2 | 0.922±5.9e-4 | 0.972±3.7e-4 | 0.748±5.0e-4 |
| ResNet-T | 0.657±9.0e-3 | 0.859±2.9e-3 | 0.441±3.2e-3 | 3.151±5.9e-2 | 0.866±1.8e-3 | 0.817±1.7e-3 | 0.724±2.0e-3 | 5.781±4.1e-2 | 0.923±6.0e-4 | 0.970±1.1e-3 | 0.749±7.8e-4 |
| ResNet-T-LR | 0.650±1.2e-2 | 0.861±2.0e-3 | 0.438±2.9e-3 | 3.163±6.1e-2 | 0.870±1.5e-3 | 0.813±2.5e-3 | 0.725±1.6e-3 | 5.687±5.9e-2 | 0.922±8.1e-4 | 0.972±3.7e-4 | 0.748±6.1e-4 |
| ResNet-P | 0.630±1.8e-2 | 0.858±3.1e-3 | 0.471±6.5e-3 | 3.147±2.9e-2 | 0.866±1.7e-3 | 0.812±1.6e-3 | 0.729±7.0e-4 | 5.566±7.5e-2 | 0.922±6.7e-4 | 0.968±7.7e-4 | 0.747±6.3e-4 |
| ResNet-PLR | 0.651±1.3e-2 | 0.859±3.7e-3 | 0.461±4.2e-3 | 3.188±7.3e-2 | 0.869±1.7e-3 | 0.816±2.5e-3 | 0.728±1.8e-3 | 5.582±4.9e-2 | 0.923±5.9e-4 | 0.972±5.1e-4 | 0.747±6.4e-4 |
| Transformer-L | 0.632±2.0e-2 | 0.860±3.0e-3 | 0.465±4.8e-3 | 3.239±3.2e-2 | 0.858±1.3e-3 | 0.817±2.3e-3 | 0.725±3.2e-3 | 5.602±4.8e-2 | 0.924±4.4e-4 | 0.971±6.8e-4 | 0.746±5.7e-4 |
| Transformer-LR | 0.614±4.5e-2 | 0.860±2.2e-3 | 0.456±3.7e-3 | 3.261±5.6e-2 | 0.858±1.6e-3 | 0.817±2.2e-3 | 0.729±1.5e-3 | 5.644±5.5e-2 | 0.924±3.9e-4 | 0.971±7.6e-4 | 0.746±5.8e-4 |
| Transformer-Q-L | 0.659±8.7e-3 | 0.856±5.9e-3 | 0.451±5.4e-3 | 3.319±4.2e-2 | 0.867±1.6e-3 | 0.812±2.6e-3 | 0.729±2.9e-3 | 5.741±4.5e-2 | 0.924±3.8e-4 | 0.973±6.1e-4 | 0.747±7.9e-4 |
| Transformer-Q-LR | 0.659±1.2e-2 | 0.857±2.0e-3 | 0.448±6.1e-3 | 3.270±4.6e-2 | 0.871±1.1e-3 | 0.812±2.5e-3 | 0.723±3.3e-3 | 5.683±4.8e-2 | 0.923±5.8e-4 | 0.972±4.2e-4 | 0.748±7.7e-4 |
| Transformer-T-L | 0.663±7.4e-3 | 0.861±1.4e-3 | 0.454±4.7e-3 | 3.197±2.9e-2 | 0.871±1.4e-3 | 0.817±2.6e-3 | 0.726±1.7e-3 | 5.803±6.5e-2 | 0.924±3.3e-4 | 0.974±4.5e-4 | 0.747±7.5e-4 |
| Transformer-T-LR | 0.665±6.6e-3 | 0.860±3.4e-3 | 0.442±5.3e-3 | 3.219±3.2e-2 | 0.870±1.5e-3 | 0.818±2.6e-3 | 0.729±1.4e-3 | 5.699±6.7e-2 | 0.924±4.4e-4 | 0.973±5.6e-4 | 0.747±8.4e-4 |
| Transformer-PLR | 0.646±2.0e-2 | 0.863±2.7e-3 | 0.464±2.8e-3 | 3.162±4.2e-2 | 0.870±1.5e-3 | 0.814±2.1e-3 | 0.730±1.9e-3 | 5.760±1.1e-1 | 0.924±5.2e-4 | 0.972±1.1e-3 | 0.746±5.9e-4 |

Table 19: Extended results for ensembles

| | GE↑ | CH↑ | CA↓ | HO↓ | AD↑ | OT↑ | HI↑ | FB↓ | SA↑ | CO↑ | MI↓ |
|---|---|---|---|---|---|---|---|---|---|---|---|
| CatBoost | 0.692±1.9e-3 | 0.861±2.4e-4 | 0.430±1.1e-3 | 3.093±5.1e-3 | 0.873±5.1e-4 | 0.825±4.7e-4 | 0.727±3.6e-4 | 5.226±1.3e-2 | 0.924±1.0e-4 | 0.967±1.4e-4 | 0.741±1.4e-4 |
| XGBoost | 0.683±1.3e-3 | 0.859±2.4e-4 | 0.434±7.1e-4 | 3.152±1.2e-3 | 0.875±5.5e-4 | 0.827±8.4e-4 | 0.726±8.1e-4 | 5.338±1.9e-2 | 0.919±4.8e-4 | 0.969±8.8e-5 | 0.742±5.3e-5 |
| MLP | 0.665±2.7e-3 | 0.856±1.2e-3 | 0.486±7.8e-4 | 3.109±1.0e-2 | 0.856±4.6e-4 | 0.822±8.0e-4 | 0.727±1.7e-3 | 5.616±7.6e-3 | 0.913±8.2e-5 | 0.968±4.8e-4 | 0.746±1.1e-4 |
| MLP-L | 0.670±3.2e-3 | 0.862±1.5e-3 | 0.471±4.7e-4 | 3.021±1.1e-2 | 0.857±5.9e-4 | 0.824±1.1e-3 | 0.728±2.7e-4 | 5.508±2.1e-2 | 0.916±1.5e-4 | 0.971±6.8e-5 | 0.746±2.3e-4 |
| MLP-LR | 0.679±4.9e-3 | 0.861±9.4e-4 | 0.463±1.9e-3 | 3.012±1.8e-3 | 0.859±8.0e-4 | 0.826±1.6e-3 | 0.731±1.1e-3 | 5.477±3.6e-2 | 0.924±7.1e-5 | 0.972±7.6e-5 | 0.744±1.6e-4 |
| MLP-LRLR | 0.676±4.8e-3 | 0.863±1.4e-3 | 0.453±1.1e-3 | 3.017±1.1e-2 | 0.858±1.6e-4 | 0.828±1.3e-3 | 0.725±5.9e-4 | 5.427±2.1e-2 | 0.924±1.2e-4 | 0.973±1.8e-4 | 0.744±1.8e-4 |
| MLP-Q | 0.677±4.8e-3 | 0.856±1.4e-3 | 0.458±1.7e-4 | 3.080±1.5e-2 | 0.862±4.0e-4 | 0.822±1.7e-3 | 0.723±5.6e-4 | 5.706±1.9e-2 | 0.922±1.7e-4 | 0.973±2.1e-4 | 0.748±2.2e-4 |
| MLP-Q-LR | 0.682±3.9e-3 | 0.859±4.7e-4 | 0.433±1.9e-3 | 3.080±9.7e-3 | 0.867±4.2e-4 | 0.818±1.4e-3 | 0.724±3.2e-4 | 5.144±1.4e-2 | 0.924±3.7e-4 | 0.974±1.5e-4 | 0.745±2.8e-4 |
| MLP-Q-LRLR | 0.674±2.9e-3 | 0.862±1.5e-3 | 0.438±2.1e-3 | 3.066±9.9e-3 | 0.870±4.1e-4 | 0.817±2.4e-3 | 0.727±2.1e-4 | 5.268±7.5e-2 | 0.924±3.1e-5 | 0.973±2.8e-4 | 0.745±1.7e-4 |
| MLP-T | 0.669±4.3e-3 | 0.861±1.0e-3 | 0.439±2.1e-4 | 3.058±1.4e-2 | 0.865±5.3e-4 | 0.822±6.3e-4 | 0.724±7.2e-4 | 5.507±2.0e-2 | 0.923±8.5e-5 | 0.972±2.7e-4 | 0.745±1.7e-4 |
| MLP-T-LR | 0.673±8.3e-4 | 0.861±8.5e-4 | 0.435±1.1e-3 | 3.099±2.4e-2 | 0.870±6.6e-4 | 0.821±2.6e-4 | 0.727±7.2e-4 | 5.409±6.2e-3 | 0.924±1.3e-4 | 0.973±1.3e-4 | 0.746±1.6e-4 |
| MLP-T-LRLR | 0.670±4.1e-4 | 0.860±2.5e-3 | 0.431±6.0e-4 | 3.056±2.2e-2 | 0.870±2.6e-4 | 0.826±5.0e-4 | 0.725±7.4e-4 | 5.440±1.8e-3 | 0.925±6.1e-5 | 0.973±2.2e-4 | 0.745±4.7e-4 |
| MLP-P | 0.661±6.0e-3 | 0.861±6.2e-4 | 0.473±1.1e-3 | 3.042±1.0e-2 | 0.871±1.1e-3 | 0.812±1.7e-3 | 0.725±6.2e-4 | 5.508±3.1e-2 | 0.924±5.4e-5 | 0.973±3.0e-4 | 0.745±2.1e-4 |
| MLP-PL | 0.671±6.2e-3 | 0.860±1.2e-3 | 0.456±1.3e-3 | 3.065±8.1e-3 | 0.872±6.3e-4 | 0.825±4.1e-4 | 0.730±3.5e-4 | 5.216±2.0e-2 | 0.924±1.2e-4 | 0.974±1.9e-4 | 0.744±2.1e-4 |
| MLP-PLR | 0.700±2.1e-3 | 0.858±1.6e-3 | 0.453±5.8e-4 | 2.975±6.6e-3 | 0.874±9.0e-4 | 0.830±2.4e-3 | 0.730±3.5e-4 | 5.388±1.6e-2 | 0.924±5.4e-5 | 0.975±4.8e-4 | 0.743±1.0e-4 |
| MLP-PLRLR | 0.699±9.3e-3 | 0.867±1.8e-3 | 0.448±8.3e-4 | 2.993±6.5e-3 | 0.873±4.1e-4 | 0.823±8.3e-4 | 0.729±9.1e-4 | 5.346±4.8e-2 | 0.924±2.6e-4 | 0.972±8.2e-4 | 0.743±9.9e-5 |
| MLP-AutoDis | 0.676±7.6e-3 | 0.860±1.7e-3 | 0.464±1.6e-3 | 3.132±5.7e-3 | 0.860±2.8e-4 | 0.817±2.1e-3 | 0.730±2.5e-4 | 5.580±2.2e-2 | 0.924±1.1e-4 | 0.970±3.2e-4 | — |
| MLP-DICE | 0.636±2.6e-3 | 0.859±2.0e-3 | 0.486±1.4e-3 | 3.092±1.3e-2 | 0.862±4.8e-4 | 0.784±2.3e-3 | 0.723±6.1e-4 | 5.615±8.9e-3 | 0.920±2.0e-4 | 0.969±1.4e-4 | 0.746±2.0e-4 |
| ResNet | 0.690±5.9e-3 | 0.861±1.6e-3 | 0.483±1.8e-3 | 3.081±7.8e-3 | 0.856±3.4e-4 | 0.821±1.8e-3 | 0.734±1.1e-3 | 5.482±1.1e-2 | 0.918±5.3e-4 | 0.968±4.4e-4 | 0.745±6.5e-5 |
| ResNet-L | 0.674±5.2e-3 | 0.859±6.2e-4 | 0.481±2.5e-3 | 3.025±1.8e-2 | 0.857±2.9e-4 | 0.819±1.3e-3 | 0.735±5.2e-4 | 5.522±2.4e-2 | 0.917±2.2e-4 | 0.966±5.1e-4 | 0.744±3.0e-4 |
| ResNet-LR | 0.672±6.0e-3 | 0.862±1.7e-3 | 0.450±2.2e-3 | 2.992±2.4e-4 | 0.859±4.7e-4 | 0.822±9.2e-4 | 0.733±4.2e-5 | 5.415±9.5e-5 | 0.923±7.7e-5 | 0.971±1.5e-4 | 0.743±2.1e-4 |
| ResNet-Q | 0.671±1.7e-3 | 0.862±8.2e-4 | 0.442±8.0e-4 | 3.128±9.0e-3 | 0.862±5.8e-4 | 0.816±9.4e-4 | 0.722±7.1e-4 | 5.402±3.3e-2 | 0.923±4.6e-4 | 0.974±6.3e-5 | 0.746±2.4e-4 |
| ResNet-Q-LR | 0.674±2.5e-3 | 0.859±1.8e-3 | 0.427±2.3e-3 | 3.066±2.2e-2 | 0.868±1.1e-3 | 0.815±7.1e-4 | 0.729±1.6e-3 | 5.309±4.9e-2 | 0.923±3.9e-4 | 0.976±1.2e-4 | 0.746±1.8e-4 |
| ResNet-T | 0.681±1.3e-3 | 0.861±2.1e-3 | 0.428±8.0e-4 | 3.064±3.6e-2 | 0.868±8.3e-4 | 0.823±4.0e-4 | 0.725±9.5e-4 | 5.657±1.5e-2 | 0.923±1.0e-4 | 0.973±6.0e-4 | 0.746±6.0e-4 |
| ResNet-T-LR | 0.683±6.1e-3 | 0.862±0.0e+00 | 0.425±7.4e-4 | 3.030±3.4e-2 | 0.872±7.3e-4 | 0.822±5.5e-4 | 0.731±1.1e-3 | 5.471±9.2e-3 | 0.923±5.8e-4 | 0.975±1.0e-4 | 0.744±3.3e-4 |
| ResNet-P | 0.675±4.2e-3 | 0.860±6.2e-4 | 0.453±3.1e-3 | 3.041±1.7e-2 | 0.872±1.4e-3 | 0.820±2.0e-4 | 0.733±5.0e-4 | 5.305±2.3e-2 | 0.923±3.6e-4 | 0.972±2.1e-4 | 0.744±1.7e-4 |
| ResNet-PLR | 0.691±6.3e-3 | 0.861±4.1e-4 | 0.443±1.4e-3 | 3.040±2.1e-2 | 0.874±5.0e-4 | 0.825±1.1e-3 | 0.734±6.3e-4 | 5.400±2.6e-2 | 0.924±2.9e-4 | 0.975±9.1e-5 | 0.743±4.0e-4 |
| Transformer-L | 0.668±1.3e-2 | 0.861±6.2e-4 | 0.455±1.4e-3 | 3.188±8.8e-3 | 0.860±6.5e-4 | 0.824±4.6e-4 | 0.727±1.1e-3 | 5.434±2.3e-2 | 0.924±1.1e-4 | 0.973±2.0e-4 | 0.743±2.7e-4 |
| Transformer-LR | 0.666±1.0e-3 | 0.861±4.1e-4 | 0.446±1.1e-3 | 3.193±1.6e-2 | 0.861±2.0e-4 | 0.824±1.6e-3 | 0.733±7.8e-4 | 5.430±3.0e-2 | 0.924±1.8e-4 | 0.973±1.0e-4 | 0.743±1.8e-4 |
| Transformer-Q-L | 0.704±1.5e-3 | 0.861±1.1e-3 | 0.426±1.6e-3 | 3.183±2.5e-2 | 0.869±2.7e-4 | 0.820±3.1e-3 | 0.735±1.5e-3 | 5.553±1.5e-2 | 0.925±2.8e-4 | 0.976±5.9e-5 | 0.744±2.0e-4 |
| Transformer-Q-LR | 0.690±1.9e-3 | 0.857±2.4e-4 | 0.425±1.2e-3 | 3.143±1.4e-2 | 0.868±4.9e-4 | 0.818±2.3e-3 | 0.726±1.1e-3 | 5.471±1.5e-2 | 0.924±2.0e-4 | 0.975±1.9e-4 | 0.744±3.5e-4 |
| Transformer-T-L | 0.693±6.3e-3 | 0.862±2.4e-4 | 0.439±1.0e-3 | 3.136±3.5e-3 | 0.872±1.4e-2 | 0.826±2.3e-3 | 0.731±1.6e-3 | 5.579±5.2e-4 | 0.924±4.0e-4 | 0.977±2.1e-4 | 0.743±2.4e-4 |
| Transformer-T-LR | 0.686±4.1e-3 | 0.862±1.1e-3 | 0.423±3.4e-3 | 3.149±1.4e-2 | 0.871±8.0e-4 | 0.823±2.4e-3 | 0.733±9.4e-4 | 5.515±2.0e-2 | 0.924±6.1e-5 | 0.976±9.2e-5 | 0.744±2.9e-4 |
| Transformer-PLR | 0.686±6.2e-3 | 0.864±9.4e-4 | 0.449±1.2e-3 | 3.091±1.3e-2 | 0.873±1.5e-3 | 0.823±1.7e-3 | 0.734±2.1e-4 | 5.581±6.4e-2 | 0.924±1.8e-4 | 0.975±2.2e-4 | 0.743±2.4e-4 |