# OpenReview forum: "On Embeddings for Numerical Features in Tabular Deep Learning"
_NeurIPS.cc/2022/Conference — NeurIPS 2022 Accept_

### Official Review · Reviewer_2HxT · 2022-07-11

**Rating:** 3
**Confidence:** 3
**Soundness:** 2 fair
**Presentation:** 2 fair
**Contribution:** 2 fair

**Summary:**

The paper presents methods for embedding numerical features for tabular deep learning methods. The paper describes the current best practices in the tabular deep learning literature, and then proposes two basic approaches for embedding numerical features and ways to combine and augment them. First, the authors propose Piecewise Linear Encoding (PLE), where a number is mapped to a T-dimenionsal vector by partitioning the range of values into T bins corresponding to the coordinates, where a coordinate is equal to: 1 if the number is below the respective bin, 0 if the number is above the bin, and otherwise the relative location of the number in the given bin, i.e., (x - low) / (high - low). This is effectively a continuous relaxation of treating numeric features as categorical via quantization. Different strategies for how to compute the bins are considered. The second method is based on mapping to a vector with periodic activation functions, i.e., mapping x -> [cos(w_1 * xx), sin(w_1 * x), ..., cos(w_n * xx), sin(w_n * x)], similar to prior proposal for positional embeddings. This paper proposes that the two base feature embeddings can be combined with some sequence of linear and ReLU activation layers before they are fed into the model.

For evaluating the different embedding methods, the paper considers three base models, a simple MLP, a ResNet, and a Transformers architecture. Various combinations of models and embedding methods are tested on a variety of tabular datasets. The main findings are that with well-chosen numerical feature embedding, deep learning-based approaches can nearly match or exceed the performance of Gradient Boosted Decision Trees. Moreover, it is shown that numerical feature embeddings are important for all model types, and even an MLP can perform exceptionally well.

**Questions:**

Could the authors explain which method is preferred and under what circumstances? Why would one method perform better on one dataset than on another, or is it all just statistical noise? I would strongly advise to move the bulk of the various variants tested to the appendix and keep only the one or two leading variants (their naming scheme is also quite confusing). It could be worthwhile to display more varied results in the context of an ablation study.

Could the authors please explain their methodology for picking the datasets to evaluate their methods? What does "GBDT-friendly dataset" mean?

I would also suggest the authors to include in their related works section a paragraph on the related methods using the NLP literature for embedding numbers.

As a very minor complaint, I would also recommend the authors ensure each time initials are introduced that their full meanings are also included. For example, GBDT appears in the abstract without explanation. CTR appears in the body without ever explaining what it stands for (Click-through Rate I guess), as well as DL (Deep Learning). Moreover, I would suggest the authors to refrain from using "dramatic" wordings (e.g., "game changer", "epic difference", "we can't help") -- let your results speak for themselves.

**Limitations:**

The authors discussed the runtime costs associated with using deep learning-based methods relative to decision trees, as well as the difficulty in drawing far reaching conclusions based on the selection of datasets and their sample size.

**Strengths And Weaknesses:**

The main strengths of the paper are its comprehensive set of experiments, including not just one variant but many possible combinations of the proposed embeddings. Such a detailed set of results could prove useful for practitioners that might consider adopting one of the proposed methods based on similarity to one of the given datasets in the evaluation. Moreover, tabular deep learning is still a relatively underexplored subject, and any push to establish better baselines could prove important.

However, the paper ignores the literature on embedding numerical features found in other domains but effectively addresses the same fundamental problem. Namely, there are many papers on how to treat numerical data found in natural language, and specifically how to embed it in the same space as conventional word embeddings. Many suggestions have been proposed over the years, some are identical to the methods in this paper, e.g., the use of periodic activation functions. While the authors note that it was proposed for positional embeddings, it was also proposed for embedding plain numbers found in the text. See [1] for a survey on the various methods used in natural language processing.

It is also difficult to discern what the bottom line is from the many experimental results, regardless of whether the proposed embedding methods are novel or not. It is unclear which method is preferred (if any), and why one method would work better than another on a given dataset. There is little attempt to analyze the results or draw any conclusions. The paper would be significantly improved if it was concentrated on one specific method rather than all possible combinations of a set of methods.

Finally, it is unclear why the paper uses the specific datasets and not others. The only explanation given is that these datasets are "GBDT-friendly". However, it is unclear what are the qualifications of a dataset to be considered as "GBDT-friendly", and why we should ignore other datasets that were previously explored in related literature on tabular deep learning. For instance, in [2] a different set of datasets is considered (though there is some overlap), where they already showed that deep learning methods can beat in most cases GBDT. In my view, this submission lacks a thorough explanation of their methodology for selecting the datasets they analyze and why it is okay to ignore others that have appeared in other papers.

[1] - Thawani et al., Representing Numbers in NLP: a Survey and a Vision, NAACL 2021.

[2] - Gorishniy et al., Revisiting Deep Learning Models for Tabular Data, NeurIPS 2021.

---

> ### Author Response · Authors · 2022-08-02
> **Response to Reviewer 2HxT (Part 2)**
>
> > However, the paper ignores the literature on embedding numerical features
>
> We thank the reviewer for pointing to NLP-related works, but we disagree with the general sentiment. In our work, we refer to classical machine learning literature, click-through rate prediction tasks, natural language processing, computer vision, implicit neural representations.
>
> Potentially relevant approaches from NLP include only so-called real-based approaches. Below, we cover all of them from [1] and explain how they are related to our work:
>
> - the only relevant piece from [2] is "value embedding", which replaces string representations with the corresponding scalar number, i.e. in our notation, this approach is the "no embeddings" approach
> - [3] and [4] are based on the NLP-specific motivation to improve learning about scales from natural text
> - the method proposed in [5] is similar in its spirit to what is called "one-blob encoding" in our ablation study (see Table 8).
> - finally, DICE [6] is the most relevant method, which embeds numbers without NLP in mind. It looks like an instance of the same idea as behind PLE (piecewise linear encoding), namely, DICE explicitly enforces some notion of order between embeddings. However, PLE with its adaptive bins is more flexible and tailored toward tabular data setups.
>
> As a sanity check, we evaluated DICE [6] on most of our datasets under the same protocol as in our paper and compare it with the pure piecewise linear encoding (since they play in the same league of non-differentiable approaches).
>
> | | GE       | CH            | CA            | HO            | AD            | OT            | HI            | FB            |
> | ---| -------- | ------------- | ------------- | ------------- | ------------- | ------------- | ------------- | ------------- |
> | MLP      | 0.632 (0.015) | 0.856 (0.003) | 0.495 (0.004) | 3.204 (0.042) | 0.854 (0.002) | 0.818 (0.003) | 0.720 (0.002) | 5.686 (0.049) |
> | MLP-DICE | 0.610 (0.013) | 0.858 (0.003) | 0.491 (0.003) | 3.146 (0.036) | 0.860 (0.001) | 0.778 (0.005) | 0.720 (0.001) | 5.726 (0.037) |
> | MLP-Q    | 0.653 (0.009) | 0.854 (0.003) | 0.464 (0.003) | 3.163 (0.032) | 0.859 (0.002) | 0.816 (0.003) | 0.721 (0.001) | 5.766 (0.055) |
> | MLP-T    | 0.647 (0.006) | 0.861 (0.001) | 0.447 (0.002) | 3.149 (0.054) | 0.864 (0.001) | 0.821 (0.002) | 0.720 (0.002) | 5.577 (0.039) |
>
> As expected, the results indicate that DICE is a suboptimal solution in the context of tabular data problems. Plus, DICE does not provide a principled mechanism for handling out-of-range values.
>
> > I would also suggest the authors to include in their related works section a paragraph on the related methods using the NLP literature for embedding numbers.
>
> We thank the reviewer for the suggestion, for now, we cite [6] in the new revision, and we are still thinking about how to incorporate literature on NLP.
>
> > Many suggestions have been proposed over the years, some are identical to the methods in this paper
>
> Unfortunately, this is hard to address without citations, we are not aware of methods identical to ours.
>
> > As a very minor complaint
>
> Thanks for noting this, we fixed these issues in the new revision.
>
> References:
>
> - [1] Avijit Thawani, Jay Pujara, Filip Ilievski, Pedro Szekely. 2021. "Representing Numbers in NLP: a Survey and a Vision".
> - [2] Eric Wallace, Yizhong Wang, Sujian Li, Sameer Singh, and Matt Gardner. 2019. "Do NLP models know numbers? probing numeracy in embeddings".
> - [3] Xikun Zhang, Deepak Ramachandran, Ian Tenney, Yanai Elazar, Dan Roth. 2020. "Do Language Embeddings capture Scales?".
> - [4] Zhihua Jin, Xin Jiang, Xingbo Wang, Qun Liu, Yong Wang, Xiaozhe Ren, Huamin Qu. 2021. "NumGPT: Improving Numeracy Ability of Generative Pre-trained Models".
> - [5] Chengyue Jiang, Zhonglin Nian, Kaihao Guo, Shanbo Chu, Yinggong Zhao, Libin Shen, and Kewei Tu. 2020. "Learning numeral embedding".
> - [6] Dhanasekar Sundararaman, Shijing Si, Vivek Subramanian, Guoyin Wang, Devamanyu Hazarika, Lawrence Carin. 2020. "Methods for Numeracy-Preserving Word Embeddings".
> - [7] Y. Gorishniy, I. Rubachev, V. Khrulkov, and A. Babenko. “Revisiting deep learning models for tabular data”. In NeurIPS, 2021

---

> ### Author Response · Authors · 2022-08-02
> **Response to Reviewer 2HxT (Part 1)**
>
> We thank the reviewer for their detailed comments.
>
> > what the bottom line is from the many experimental results ... Could the authors explain which method is preferred ...
>
> Please, see the global reply, where we describe the relationship between the methods and give practical recommendations.
>
> > What does "GBDT-friendly dataset" mean?
>
> Thanks for this question, we clarified that in the new revision. The dataset is GBDT-friendly if GBDT outperforms models without embeddings such as MLP and ResNet (please, see Table 6; also, see L285-286).
>
> > Finally, it is unclear why the paper uses the specific datasets and not others. ... why we should ignore other datasets ... Could the authors please explain their methodology for picking the datasets to evaluate their methods?
>
> We already describe how we selected datasets in section 4.1. Please note that the field still lacks a standard ImageNet/GLUE-like benchmark. The best thing we can do is to pick a reasonable number of datasets with a reasonable motivation and transparently communicate it.
>
> Regarding [7]:
>
> - we actually use it as a starting point for collecting datasets
> - Table 4 from that paper indicates that both lightweight and heavy DL models already outperform GBDT on the datasets HE, JA, AL, EP, YE. In the context of "DL vs GBDT" competition, there was little motivation to apply embeddings on problems where lightweight models are already enough.
> - That said, we still include non-GBDT-friendly datasets in our paper to mitigate the bias (e.g. see HO and HI in Table 6 in our paper)
>
> > For instance, in ... they already showed that deep learning methods can beat in most cases GBDT
>
> We disagree, since the abstract of that paper says "there is still no universally superior solution" implying that "DL vs GBDT" is an open question. Moreover, the competitive results in that paper are achieved with the heavy attention-based model, while we managed to design competitive MLP-like models.

---

### Official Review · Reviewer_5oG4 · 2022-07-11

**Rating:** 5
**Confidence:** 3
**Soundness:** 3 good
**Presentation:** 3 good
**Contribution:** 3 good

**Summary:**

This paper address tabular data prediction by using deep models. The authors argue that embedding for numerical features are critical but underexplored in previous work. In practice, two embedding modules termed piecewise linear encoding and periodic activations are proposed to improve the performance of the current deep models for tabular data. Extensive experimental results demonstrate the effectiveness of the proposed two embedding schemes.

**Questions:**

See the weaknesses. (1)The motivation of the module design is still not clear; (2) How about the performance gain of the proposed two modules for more recent proposed deep architecture.

**Limitations:**

The authors partially discussed the limitations of the current work in the conclusion and future work.

**Strengths And Weaknesses:**

Strengths
+ The paper is clearly written and easy to follow
+ The proposed two embedding schemes are straightforward and have practical effectiveness.
+ Extensive experimental results illustrate the performance of the proposed method on various deep model architectures.

Weaknesses
- What’s the insight of designing such two embedding modules? It is more like the practical solutions that depend on massive experiments, which makes me wonder about the intuition behind these choices.

- What’s the relationship between these two proposed embedding modules? Should we need to consider them as independent contributions, and why? Are there any different application scenarios for these two modules? Which one is more effective in real applications?

- Is there any more deep architectures for tubular data analysis, and how about their performance by adopting the proposed two modules. If the authors aim to demonstrate the generality of the proposed module, more recent proposed deep networks (but not the simple Resnet or Transformer) should be considered as the baseline models, and their performance improvement by using proposed modules should be clearly reported.

---

> ### Author Response · Authors · 2022-08-02
> **R3 answer**
>
> We thank the reviewer for the review.
>
> > What’s the insight of designing such two embedding modules?
>
> In fact, we motivate our choices throughout the paper:
>
> - First, in the first paragraph of section 3.2, we take motivation from [1] and [2] to start the exploration of alternative ways of dealing with numerical features
> - Then, in the next paragraph of the same section, we take inspiration from one-hot encoding to develop its continuous variation for numerical features. Another perspective is that Piecewise Linear Embeddings are a natural evolution of simple linear embeddings which already showed their effectiveness in [3].
> - Finally, in section 3.3, we turn to [2] once again to check whether it is possible to adapt the proposed method for tabular data problems (importantly, in the vanilla form, their method does not work well, which we show in Table 11).
>
> And it turns out that the described methods are already enough to illustrate the usefulness of embeddings for tabular problems and propose specific practical approaches.
>
> > What’s the relationship between these two proposed embedding modules? Which one is more effective in real applications?
>
> Please, see the global reply, where we describe the relationship between the methods and give practical recommendations.
>
> > If the authors aim to demonstrate the generality of the proposed module, more recent proposed deep networks (but not the simple Resnet or Transformer) should be considered
>
> Please, see the paragraph just before Table 6, where we explain our choice of the backbones. We believe that our analysis covers all essential architectures, since [3] effectively demonstrated that the MLP, ResNet and Transformer backbones form the representative spectrum of what deep learning backbones are capable of, and what practitioners and researchers are most likely to use in practice. Moreover, even this set of models already required huge compute for proper analysis, so adding more backbones would be both very expensive and, perhaps, not that valuable for users.
>
> References:
>
> - [1] N. Rahaman, A. Baratin, D. Arpit, F. Draxler, M. Lin, F. A. Hamprecht, Y. Bengio, and A. C. Courville. "On the spectral bias of neural networks". In ICML, 2019.
> - [2] M. Tancik, P. P. Srinivasan, B. Mildenhall, S. Fridovich-Keil, N. Raghavan, U. Singhal, R. Ramamoorthi, J. T. Barron, and R. Ng. "Fourier features let networks learn high frequency functions in low dimensional domains". In NeurIPS, 2020.
> - [3] Y. Gorishniy, I. Rubachev, V. Khrulkov, and A. Babenko. "Revisiting deep learning models for tabular data". In NeurIPS, 2021

---

> ### Author Response · Authors · 2022-08-05
> **Dear Reviewer 5oG4**
>
> Dear Reviewer 5oG4,
>
> Does our reply below ("R3 answer") properly answer your questions? If you have more comments and feedback, we will be glad to continue the discussion.

---

### Official Review · Reviewer_GNk8 · 2022-07-12

**Rating:** 6
**Confidence:** 4
**Soundness:** 3 good
**Presentation:** 3 good
**Contribution:** 2 fair

**Summary:**

This paper proposes a new encoding scheme for embedding numerical features in neural network models. The authors report that for tabular data in many cases neural methods have failed to improve on Gradient Boosted Decision Trees (GBDT), thus necessitating more research into how to represent the tabular data, in particular features that are numerical as opposed to categorical. The authors propose a variation of a binning/bucketing scheme which they call Piecewise Linear Encoding (PLE), which is more informative than a one-hot encoding scheme, where a numerical feature is simply represented by the embedding corresponding to the bin/bucket id it belongs to. This is clearly a bad scheme for numerical features, since the ordering information is lost: if x_1 > x_2, then there is no ordering relationship between the embedding(x_1) and embedding(x_2), so the model must first learn this implicit relationship. Such a scheme therefore loses some information in the case of numerical data. The authors propose their PLE scheme as follows: for the data x, if it falls between bin b_t and b_{t + 1}, then embedding_i = 0 if i < t, embedding_i = 1 if i > t, and embedding_t = (x - b_t)/(b_{t+1} - b_t). Thus this is a piece-wise linear scheme which explains the naming of the scheme. The authors also consider some variations inspired by the position encoding used in Transformer models for text, where embedding_i also has a sinusoidal component.

The authors show experimental results using this scheme where they show that on several tabular data-sets this scheme improves on the baseline MLP models using simple linear embeddings and is competitive with decision trees.


**Questions:**

Comments:
- Line 14: "Specifically, after proper embeddings, simple MLP-like models can perform on par with the attention-based architectures"

- Line 44: "In particular, we show, that after proper embeddings, simple MLP-like architectures often provide the performance comparable to the state-of-the-art attention-based models."

The statements in line 14 and line 44 should be qualified with a "for numerical data", since that is the correct scope of the claim as presented in this work.

- Line 55 "3. On a number of public benchmarks, we achieve the new state-of-the-art of tabular DL." -> "state-of-the-art on tabular DL" Typo, please fix.

References:

The piecewise encoding scheme is quite similar to the "thermometer encoding scheme" proposed in [1] for representing image intensities, with the main difference being the absence of the piece-wise linear term (x - b_t)/(b_{t+1} - b_t). The motivation was to see whether this representation conferred any adversarial robustness (which it did not), but improvements in standard (vanilla) accuracy of ResNets was observed using this scheme on MNIST, CIFAR-10 and SVHN datasets (see Table 2, 3, 4 corresponding to "vanilla" and "clean").

[1] @inproceedings{buckman2018thermometer,
  title={Thermometer encoding: One hot way to resist adversarial examples},
  author={Buckman, Jacob and Roy, Aurko and Raffel, Colin and Goodfellow, Ian},
  booktitle={International Conference on Learning Representations},
  year={2018}
}

Suggestions: It would make the work more convincing if the authors reported results on larger CTR datasets like CRITEO.

**Limitations:**

Yes

**Strengths And Weaknesses:**

Strengths: This paper explores a few different representation schemes to make neural networks work better for tabular data and present empirical results demonstrating that it improves on other simpler representation schemes such as linear or one-hot encoding schemes. The paper is also clearly written and the notation is easy to follow.

Weaknesses: Several of the data-sets used are likely very small scale so it is possible that in the abundant data regime the gains from this method are small. I would also imagine that the reason the decision trees are better than neural networks on tabular data of the kind the authors study is also likely because it seems like a data limited setting. In large(r) CTR datasets like CRITEO, it is not clear that 1) decision trees would be better than MLPs and 2) the encoding scheme would give as big of a "win".

---

> ### Author Response · Authors · 2022-08-02
> **Response to Reviewer GNk8 (Part 2)**
>
> > Several of the data-sets used are likely very small scale so it is possible that in the abundant data regime the gains from this method are small. I would also imagine that the reason the decision trees are better than neural networks on tabular data of the kind the authors study is also likely because it seems like a data limited setting. In large(r) CTR datasets like CRITEO, it is not clear that 1) decision trees would be better than MLPs and 2) the encoding scheme would give as big of a "win".
>
> Overall, in our benchmarks, there are already not-so-small datasets with 500K+ objects (see Table 1), so we believe that the observed effect is not too niche, is applicable to considered dataset sizes, and may generalize to other sizes due to our simple training and tuning protocols (no fancy optimizers, lr schedulers, augmentations etc.).
>
> Nevertheless, below, we:
>
> - **(A)** run a simple experiment to study how the effect of embeddings depends on the dataset size
> - **(B)** explain why dataset size may not be the key property defining the effect from embeddings
>
> **(A)**
>
> (**Disclaimer:** unfortunately, due to limited time, we could not run our methods and protocols on huge datasets. We realize that the reviewer considers our dataset as already small, but we did the best we could.)
>
> We run the following experiment. For one dataset, we gradually reduce its size and measure the performance of MLP and MLP-PLR (the model with the best rank in our paper). This way, we try to check whether the rule "the less data - the better for embeddings" works. The tuning and evaluation protocols are the same as in the paper.
>
> | Model (dataset fraction)    | CA (RMSE) | HO (RMSE) | HI (Accuracy)|
> | --- | --------- | ----------| -------------|
> | MLP (0.125) |  0.589 (0.004) | 3.567 (0.101) | 0.690 (0.003) |
> | MLP (0.25) |  0.570 (0.004) | 3.391 (0.059) | 0.696 (0.002) |
> | MLP (0.5) |  0.528 (0.004) | 3.339 (0.067) | 0.709 (0.002) |
> | MLP (1.0) |  0.495 (0.004) | 3.204 (0.042) | 0.720 (0.002) |
> | MLP-PLR (0.125) |  0.557 (0.008) | 3.636 (0.098) | 0.703 (0.002) |
> | MLP-PLR (0.25) |  0.522 (0.007) | 3.373 (0.091) | 0.710 (0.003) |
> | MLP-PLR (0.5) |  0.493 (0.005) | 3.194 (0.073) | 0.720 (0.002) |
> | MLP-PLR (1.0) |  0.467 (0.006) | 3.050 (0.035) | 0.728 (0.002) |
>
> Overall, it looks like it depends on a dataset. On the CA and HI datasets, performance changes are similar between the vanilla model and the model with embeddings. On the HO dataset, however, it looks like the model with embeddings cannot be trained properly when data is limited and it becomes comparable to the embedding-free model at the 0.25 subsampling rate and becomes even worse at the 0.125 subsampling rate.
>
> **(B)**
>
> While dataset sizes may correlate with GBDT vs DL results `[3]`, this may not be the defining dataset property. Please, see the first paragraph of section 3.2, where we take inspiration from `[1]` and `[2]` to start our exploration of alternative numerical feature processing strategies.
>
> The main point is that there are other properties that can prevent a simple ReLU-based MLP from learning the correct dependencies even when **infinite** amount of data is available. The simplest example is the "checkboard" problem, where a model is to solve a binary classification problem (e.g. predict the color of the cell) from its two coordinates. The ReLU-based MLP will always yield blurry predictions on this task, even though an infinite amount of training data points can be generated. So other fixes are needed (e.g. embeddings or smart activations).
>
> References:
>
> - `[1]` N. Rahaman, A. Baratin, D. Arpit, F. Draxler, M. Lin, F. A. Hamprecht, Y. Bengio, and A. C. Courville. "On the spectral bias of neural networks". In ICML, 2019.
> - `[2]` M. Tancik, P. P. Srinivasan, B. Mildenhall, S. Fridovich-Keil, N. Raghavan, U. Singhal, R. Ramamoorthi, J. T. Barron, and R. Ng. "Fourier features let networks learn high frequency functions in low dimensional domains". In NeurIPS, 2020.
> - `[3]` Léo Grinsztajn, Edouard Oyallon, Gaël Varoquaux "Why do tree-based models still outperform deep learning on tabular data?"

---

> > ### Author Response · Authors · 2022-08-02
> > **Response to Reviewer GNk8 (Part 1)**
> >
> > We thank the reviewer for the detailed feedback.
> >
> > > The statements in line 14 and line 44 should be qualified with a "for numerical data"
> >
> > Thanks for noting this. Could you, please, elaborate a bit more on how we can improve the formulations? Embeddings for numerical features imply the presence of numerical features. The "amount" of numerical features is also not an easy question, for example, on the Adult ("AD" in the paper) dataset, embeddings bring much benefits, while there are only 6 numerical features (and 8 categorical features).
> >
> > > Typo, please fix
> >
> > Thanks, we fixed this in the new revision.
> >
> > > The piecewise encoding scheme is quite similar to the "thermometer encoding scheme"
> >
> > We already compared with this encoding in Table 8, but we called it "binary", and now we renamed it to "thermometer" in the new revision, thanks for pointing to this work. It looks like this work focuses on the effect of this transformation specifically in the computer vision domain, because in its details (lossy encoding, uniformly chosen bins) the scheme may be too simple for tabular data.

---

### Official Review · Reviewer_HiMf · 2022-07-13

**Rating:** 6
**Confidence:** 4
**Soundness:** 3 good
**Presentation:** 4 excellent
**Contribution:** 3 good

**Summary:**

The paper proposes several approaches for numerical feature embeddings for tabular deep learning models and experimentally investigates the gains that such approaches provide. Methodologically, simple linear embeddings, piece-wise linear embeddings, and periodic-activation-based embeddings are proposed and experimentally evaluated on a suite of many benchmark tabular datasets. The experimental results showcase an improved performance competitive with gradient boosted decision trees (GBDT) on GBDT-biased datasets.

**Questions:**

* Line 286: "However, after coupling DL models with embeddings for numerical features, the situation changes and the MI dataset remains the only one where all DL models lag behind GBDT." It is true that all deep models lag behind only on a single dataset. However, this seems like a one-to-many comparison. Do you have recommendations for a single best model or a good rule of thumb? What should a practitioner use?

* Table 6: Simple MLP achieves the best performance (with PLR embeddings). In general, from Table 6 we see that MLP>ResNet>Transformer when numerical embeddings are used. Do you have any explanation/intuition about why?



**Limitations:**

The authors stated that the work focuses on a generic aspect of deep learning models and that therefore the negative societal impact discussion does not apply. The limitations have been adequately addressed.

**Strengths And Weaknesses:**

Strengths:
* Possible performance improvements from using embeddings for numerical features are showcased
* The experiments are conducted on a large suite of datasets
* Good average rank improvements over GBDT on a GBDT-friendly benchmark are achieved by using numerical embeddings
* The paper is very well-written and easy-to-follow
* The code is in good shape

Weaknesses:
* These embeddings for numerical features are essentially feature engineering. Comparison to GBDT with feature engineering would dramatically improve the paper and the experimental results.
* Methodological contributions of piece-wise linear embeddings are similar to feature binning [1] as the paper mentions in the introduction.
* I believe automated feature engineering methods such as e.g. [2,3,4] should be mentioned in related work.
* While the proposed numerical feature embedding approaches are compared to each other, it is unclear if they provide any benefit over e.g. SAINT [5] with their dense layer or small MLP embeddings.


[1] Dougherty, J., Kohavi, R. and Sahami, M., 1995. Supervised and unsupervised discretization of continuous features. In Machine learning proceedings 1995 (pp. 194-202). Morgan Kaufmann.

[2] Khurana, U., Samulowitz, H. and Turaga, D., 2018, April. Feature engineering for predictive modeling using reinforcement learning. In Proceedings of the AAAI Conference on Artificial Intelligence (Vol. 32, No. 1).

[3] Kaul, A., Maheshwary, S. and Pudi, V., 2017, November. Autolearn—Automated feature generation and selection. In 2017 IEEE International Conference on data mining (ICDM) (pp. 217-226). IEEE.

[4] Khurana, U., Turaga, D., Samulowitz, H. and Parthasrathy, S., 2016, December. Cognito: Automated feature engineering for supervised learning. In 2016 IEEE 16th International Conference on Data Mining Workshops (ICDMW) (pp. 1304-1307). IEEE.

[5] Somepalli, G., Goldblum, M., Schwarzschild, A., Bruss, C.B. and Goldstein, T., 2021. Saint: Improved neural networks for tabular data via row attention and contrastive pre-training. arXiv preprint arXiv:2106.01342.

---

> ### Author Response · Authors · 2022-08-02
> **Response to Reviewer HiMf (Part 2)**
>
> > These embeddings for numerical features are essentially feature engineering. Comparison to GBDT with feature engineering would dramatically improve the paper and the experimental results. ... feature engineering methods ... should be mentioned in related work.
>
> Below, we do two things:
>
> - **(A)** out of curiosity, we evaluate XGBoost with feature engineering based on periodic functions
> - **(B)** we explain why the "feature engineering" perspective, while semantically interesting, is not fully accurate in our case
>
> **(A)**
>
> In the table below,  we report results for XGBoost with feature engineering. The datasets and protocols are exactly the same as in the paper. When feature engineering is used, we add new features *and keep the original features* (it seems to be important for XGBoost). The new features are obtained from the `Periodic` module as defined in `Equation (2)` in the paper, but the `c_i` coefficients are random (since there is no end-to-end training).
>
> |                    | CA (RMSE)     | HO (RMSE)     | HI (Accuracy) |
> |--------------------|---------------|---------------|---------------|
> | XGBoost            | 0.436 (0.003) | 3.160 (0.007) | 0.724 (0.001) |
> | XGBoost + Periodic | 0.441 (0.003) | 3.184 (0.015) | 0.724 (0.002) |
> | MLP                | 0.495 (0.004) | 3.204 (0.042) | 0.720 (0.002) |
> | MLP-PL             | 0.467 (0.003) | 3.113 (0.033) | 0.727 (0.002) |
>
> The table indicates that this type of feature engineering is not that beneficial for XGBoost (deep models get profit from embeddings on these datasets).
>
> **(B)**
>
> We assume that the reviewer implies "automatic" feature engineering since manual feature engineering in its ultimate form can make even linear models provide state-of-the-art results.
>
> **First**, if a feature engineering algorithm is not trained end-to-end with the whole model then it is an orthogonal component universally applicable to GBDT, neural networks, and all other models. Hence, we do not consider those, as well as all other techniques capable of improving any ML model.
>
> **Second**, our methods are not supposed to generate features for other models. Adding the obtained embeddings as new features to GBDT is definitely possible. But it looks like a suboptimal strategy (also see the experiment above), and, most importantly, then we enter a different league of multi-model approaches with its own rules, baselines and technical complexity. Namely, we would have to:
>
> - compare to stacking (it can be better and simpler to add a single prediction as a new feature instead of all embeddings as many new features)
> - compare to penultimate embeddings (it can be better and simpler to add a moderate number of new features prediction as a new feature instead of all embeddings as many new features)?
> - perform more sophisticated data splits to avoid overfitting
> - compare with all existing feature engineering methods (including those mentioned by the reviewer)
> - etc.
>
> That said, there is one exception in the form of Piecewise Linear Encoding (PLE). Since it is used *before* the main optimization, it is not specific to deep models and it can be used with GBDT as well. However, it is a threshold-based method, so not only it cannot help GBDT, but it can harm it, because the thresholds for PLE are chosen without GBDT in mind. Also, the modern GBDT implementations use histogram-based training and perform a very similar transformation under the hood (e.g. see the `max_bin` parameter in XGBoost). Moreover, **it is included in our ablation study** under the name "thermometer" in Table 8.

---

> ### Author Response · Authors · 2022-08-02
> **Response to Reviewer HiMf (Part 1)**
>
> We thank the reviewer for the thorough review. We are glad that they appreciated the empirical results. Below, we answer the questions and address the remarks regarding "feature engineering".
>
> > It is true that all deep models lag behind only on a single dataset. However, this seems like a one-to-many comparison.
>
> Thank you for pointing to this. The point was that there is a single dataset in the benchmark where embeddings failed to fully close the gap. In the updated submission file, we use the following improved formulation on L287-290 (in addition to being more transparent, it is also a bit stronger, which is surprising even to us):
>
> *"However, for the vast majority of the “backbone & dataset” pairs, proper embeddings are the only thing needed to close the gap with GBDT. Exceptions (rather formal) include the MI dataset and the following pairs: “ResNet & GE”, “Transformer & FB”, “Transformer & GE", “Transformer \& OT”."*
>
> > Do you have recommendations for a single best model or a good rule of thumb? What should a practitioner use?
>
> Please, see the global reply, where we describe the relationship between the methods and give practical recommendations.
>
> > Table 6: Simple MLP achieves the best performance ... Do you have any explanation/intuition about why?
>
> From our benchmark, it looks like deeper and more complex representations (that ResNet and Transformer can provide) are less important than the improved handling of numerical features. Overall, it makes sense, since, in tabular data, many raw features are already high-level enough, contrary to raw pixels in images. That said, while we actively highlight the performance boosts for MLPs, we think it is a bit too early to make "final" claims about the backbones.
>
> > it is unclear if they provide any benefit over e.g. SAINT [5] with their dense layer or small MLP embeddings.
>
> In fact, we analyze embeddings similar to those from SAINT. Namely, we analyze "LR" and "LRLR" (see Table 2 and L229), while SAINT, in our notation, uses "LRL" (Linear-ReLU-Linear) in their source code.

---

### Author Response · Authors · 2022-08-02
**Global reply to the reviewers**

(R1 = Reviewer HiMf, R2 = Reviewer GNk8, R3 = Reviewer 5oG4, R4 = Reviewer 2HxT)

We thank the reviewers for the detailed reviews.
We are excited to learn that they appreciated the performance improvements provided by the described methods (R1, R2, R3), as well as the extensive experiments (R1, R3, R4). Also, we are glad to learn that the reviewers found our presentation easy to follow (R1, R2, R3).

We provide individual replies under the corresponding reviews. Also, we incorporated the feedback and updated the main submission file. The changes include:

- improved communication (improved wording as suggested by R4, clarified abbreviations as suggested by R4, fixed typos as suggested by R3)
- improved claim regarding the "DL vs GBDT" comparison as suggested by R1, see L287-290
- improved positioning of the embedding schemes. We thank the reviewers for raising questions about the relation between embeddings (R1, R3, R4), indeed, positioning could be less ambiguous. The specific changes can be found on L41-L43, L128, and the in most of the bullets in the "The main takeaways for DL models" paragraph on L268.

Regarding the last point, for convenience, we provide the less formal summary below:

- The "LR" embedding (Linear + ReLU) serves as a *sanity check* for developing more sophisticated methods since it is the simplest possible non-linear embedding scheme. The good news is that even this simple module can provide a performance boost. **Use cases**: (A) when there is no time for understanding and tuning more advanced embedding modules (B) an easy-to-use baseline tabular DL solution.
- PLE is a *simple interpretable non-differentiable encoding* in the spirit of classical ML. While it allows building well-performing embeddings (e.g. T-LR), it also allows building more lightweight embeddings (see MLP-T in Table 7), is less sensitive to preprocessing (see Table 10 in the Appendix), and overall illustrates that stacking differentiable black-box modules may not be the final answer for embeddings (also, see Figure 5 in the Appendix). **Use cases**: when there are special requirements in terms of interpretability, efficiency and simplicity.
- "PLR" embedding (Periodic + Linear + ReLU) is a powerful end-to-end trainable approach with the best results on average. It is less lightweight and requires tuning the important $\sigma$ hyperparameter. **Use cases**: when there are no special requirements in terms of interpretability and efficiency.

Finally, **regarding contributions**, we would like to highlight that one of the important contributions is the phenomenon itself, that is, the effectiveness (and sometimes truly impressive performance boosts) that embeddings for numerical features can provide for tabular DL models, especially for MLP-like models. To the best of our knowledge, this topic was not systematically studied in the context of tabular deep learning.

---

### Author Response · Authors · 2022-08-07
**Dear Reviewers**

Dear Reviewers,

We are open for further feedback and discussion.

Authors

---

### Meta-Review · Area_Chair_bwLk · 2022-08-26

**Recommendation:** Accept
**Confidence:** Less certain

**Metareview:**

The authors propose and study the use of embedding scheme to apply deep learning to tabular problems. According to reviewers HiMf and 5oG4 and reading the submission, the method is simple and clearly explained. The experiments are comprehensive and demonstrates empirical improvements on small scale datasets. Moreover, discussions with reviewers have allowed the authors to provide additional relevant experiments providing comparisons with other methods.
I recommend this paper for acceptance.

**Award:**

No

---

### Decision · Program_Chairs · 2022-09-14

Accept